# Description and Evaluation of a Secondary Organic Aerosol and New Particle Formation Scheme within TM5-MP v1.12

Tommi Bergman[1,2], Risto Makkonen[2,3], Roland Schrödner[7,5], Erik Swietlicki[4], Vaughan T. J. Phillips[6], Philippe Le Sager[1], and Twan van Noije[1]

[1]Royal Netherlands Meteorological Institute (KNMI), PO Box 201, 3730 AE De Bilt, the Netherlands
[2]Climate System Research, Finnish Meteorological Institute, P.O. Box 503, FI-00101, Helsinki, Finland
[3]Institute for Atmospheric and Earth System Research / Physics, Faculty of Science, University of Helsinki, P.O. Box 64, FI-00014, Finland
[4]Department of Physics, Lund University, Box 118, 22100 Lund, Sweden
[5]Institute for Tropospheric Research, Permoserstr. 15, 04318 Leipzig, Germany
[6]Department of Physical Geography and Ecosystem Science, Lund University, Solvegatan 12, Lund 22362, Sweden
[7]Centre for Environmental and Climate Science, Lund University, Sölvegatan 37,Lund 22362, Sweden

**Correspondence:** Tommi Bergman (tommi.bergman@fmi.fi)

**Abstract.** We have implemented and evaluated a secondary organic aerosol scheme within the chemistry transport model TM5-MP in this work. In earlier versions of TM5-MP the secondary organic aerosol was emitted as Aitken sized particle mass emulating the condensation. In the current scheme we simulate the formation of SOA from oxidation of isoprene and monoterpenes by ozone and hydroxyl radicals which produce semi-volatile organic compounds and extremely low-volatility compounds. Subsequently, SVOC and ELVOC can condense on particles. Furthermore, we have introduced a new particle formation mechanism depending on the concentration of ELVOCs. For evaluation purposes, we have simulated the year 2010 with the old and new scheme, where we see an increase in simulated production of SOA from $39.9$ $\text{Tgyr}^{-1}$ with the old scheme to $52.5$ $\text{Tgyr}^{-1}$ with the new scheme. For more detailed analysis, the particle mass and number concentrations and their influence on the simulated aerosol optical depth are compared to observations. Phenomenologically, the new particle formation scheme implemented here is able to reproduce the occurrence of observed particle formation events. However, the modelled concentrations of formed particles are clearly lower than in observations as is the subsequent growth to larger sizes. Compared to the old scheme, the new scheme is increasing the number concentrations across the observation stations while still underestimating the observations. The organic aerosol mass concentrations in the US show a much better seasonal cycle and no clear overestimation of mass concentrations anymore. In Europe the mass concentrations are lowered leading to a larger underestimation of observations. Aerosol optical depth is generally slightly increased except in the northern high latitudes. This brings the simulated annual global mean AOD closer to observational estimate. However, as the increase is rather uniform, biases tend to be reduced only in regions where the model underestimates the AOD. Furthermore, the correlations with satellite retrievals and ground-based sun-photometer observations of AOD are improved. Although the process based approach to SOA formation causes reduction in model performance in some areas, overall the new scheme improves the simulated aerosol fields.

# 1 Introduction

Aerosols have a pronounced influence on the climate (Forster et al., 2007; Boucher et al., 2013) and air quality (Isaksen et al., 2009; Monks et al., 2009). Particulate organic matter, also known as organic aerosol (OA), contributes between 20 % and 90 % of total aerosol mass (Kanakidou et al., 2005). This ubiquitous OA is a major component of the atmospheric aerosols across the globe (Zhang et al., 2007). It has two main sources, which are separated due to their formation mechanism. On the one hand, organic mass is emitted directly to the atmosphere. This component is often called primary organic aerosol (POA). On the other hand, OA is formed in the atmosphere by oxidation of gaseous organic compounds. This part is known as secondary organic aerosol (SOA). POA sources include fossil fuel combustion, biofuel burning or wildfires, while organic gases are released into the atmosphere from both natural and anthropogenic sources producing volatile organic compounds (VOC). These VOCs can undergo chemical reactions in the atmosphere producing organic compounds with lower volatilities. Furthermore, some of them can take part in new particle formation (NPF) and condense onto existing particles, thereby creating SOA. Depending on their volatility and their contribution to these two processes, these low-volatility products are often separated into lumped species. In the present model we separate low-volatility products into semi-volatile VOCs (SVOCs) and extremely low-volatility VOCs (ELVOCs).

Natural sources of VOCs account for approximately 85 % of total VOC emissions (Guenther et al., 2012; Lamarque et al., 2010). Important natural sources of VOC include e.g., terrestrial vegetation or marine phytoplankton (Guenther et al., 2006, 2012; Meskhidze and Nenes, 2006; Gantt et al., 2009; Yassaa et al., 2008; Shaw et al., 2003). Due to their biogenic origin these VOCs are often referred to as biogenic VOCs (BVOC). Emissions of BVOCs are dominated by isoprene and terpenes (Guenther et al., 2012; Glasius and Goldstein, 2016). Their contribution to OA production is significant and highlights the importance of the interactions between biosphere and atmosphere within the Earth system (Carslaw et al., 2010; Paasonen et al., 2013). Emissions of BVOCs constitute roughly 90% of the total VOC emissions, but due to complex chemistry the actual processes and the total amount of SOA formation is rather uncertain (Tsigaridis et al., 2014). The estimates of the total annual production of SOA from bottom-up and top-down methods range between 12 and 1820 $\mathrm{Tg\,yr^{-1}}$ (Goldstein and Galbally, 2007; Hallquist et al., 2009). With deficiencies in understanding and complex pathways the descriptions of SOA formation in global models are often rudimentary. In the study by Tsigaridis et al. (2014) many models still treated SOA simply by emitting it as OA with prescribed SOA mass yields. These models with prescribed yields produce SOA amounts near the lower limit of the estimated source strength. In addition, models that treat SOA formation with a simple chemistry estimate similarly low SOA production. Models with more complex treatment of chemical reactions leading to SOA formation can produce up to 121 $\mathrm{Tg\,yr^{-1}}$ of SOA (IMAGES (Müller et al., 2009; Stavrakou et al., 2009; Ceulemans et al., 2012) and IMPACT (Lin et al., 2012) models in Tsigaridis et al. (2014)). However, more intricate parameterisations that are commonly used for regional applications, such as those based on a volatility basis set (Donahue et al., 2011), require the consideration of a much larger number of species and are therefore often not feasible in global chemistry transport models (CTMs) or Earth system models (ESMs).

The partitioning of VOCs to aerosol particles has traditionally been described following the partitioning theory by Pankow (1994). Accordingly, the VOCs are treated as semi-volatile vapours, which can reach an equilibrium between the gas and the

particulate phase, which depends on the individual volatility of the considered VOC species. In this theory a fraction of the
55 gas-phase VOCs condenses onto existing aerosol particles in proportion to the OA mass they contain (Riipinen et al., 2011). This approach, however, cannot explain the growth of smaller particles. The observed growth of nanometer-sized particles can only be explained if the fraction of the VOC that condenses is proportional to the available particle surface area (Riipinen et al., 2011). The growth of existing particles due to the formation of SOA from oxidation products of isoprene and monoterpenes can increase the number concentration of cloud condensation nuclei (CCN) (e.g. Duplissy et al., 2008; Engelhart et al., 2008,
2011).

Atmospheric new particle formation (NPF) is the ubiquitously occurring formation of molecular clusters and their growth to particles tens of nanometers in diameter (Kulmala et al., 2004; Nieminen et al., 2018; Kerminen et al., 2018). NPF requires gaseous compounds of very low volatility, such as sulfuric acid or highly oxidised organic compounds. For example, it was shown that sulfuric acid in concert with organics plays a key role in the formation and early growth of particles (Smith et al.,
2008; Metzger et al., 2010; Paasonen et al., 2013; Ehn et al., 2014). However, in global models it has been very common to use parameterisations for binary homogeneous nucleation of sulfuric acid and water, which predicts the NPF reasonably well in the free troposphere but fails to reproduce the concentrations in the boundary layer (Spracklen et al., 2010; Mann et al., 2012). Due to recent advances, new parameterisations for NPF in the boundary layer were developed, e.g. involving sulfuric acid (Sihto et al., 2006), ammonia (Dunne et al., 2016) and VOCs in general (Paasonen et al., 2010; Riccobono et al., 2014;
Bergman et al., 2015). After growth due to condensation or coagulation, newly formed particles can contribute to the global CCN number budget (e.g. Wang and Penner, 2009; Merikanto et al., 2009; Makkonen et al., 2012; Dunne et al., 2016; Gordon et al., 2017; Kerminen et al., 2018).

In conclusion, SOA formation can affect both NPF and condensational growth of existing aerosol particles. The two processes can promote the growth of particles into sizes relevant for CCN and therefore affect cloud properties. However, this
effect is highly non-linear as the two processes distribute SOA differently on the aerosol size spectrum. Whereas NPF initially produces a large number of small particles, condensational growth increases the size of existing particles. Whether the addition of new particles due to SOA formation increases or decreases the number of CCN therefore depends on the share of SOA mass divided between NPF and condensation.

In this work we present the implementation of a VOC oxidation scheme to calculate the production of ELVOCs and SVOCs
to describe the formation of SOA within TM5-MP. Additionally, to improve the description of NPF in the boundary layer, we implemented a NPF scheme following Riccobono et al. (2014), which describes the production of new particles in the presence of ELVOCs. The new SOA and NPF scheme is part of the TM5 version of EC-Earth3-AerChem (Döscher et al., 2021; van Noije et al., 2021), which is used in AerChemMIP (Aerosol Chemistry Model Intercomparison Project; Collins et al., 2017) of the Coupled Model Intercomparison Project phase 6 (CMIP6; Eyring et al., 2016). In this paper we describe and evaluate
the new SOA and NPF scheme. TM5 simulations with and without the new scheme are compared and evaluated against in-situ and remote sensing datasets. The performance of the new SOA treatment is evaluated for key variables, such as SOA budget, total organic aerosol mass, aerosol optical depth (AOD) and aerosol number concentrations in the surface layer.

In Section 2 we describe the new SOA formation and NPF schemes along with the observational data used for evaluation. In Section 3 we present an evaluation of the simulation against observations. In Section 5 we give the conclusions.

## 2    Model description

### 2.1    Chemistry transport model TM5-MP

In this work we use and develop the global 3D chemical transport model TM5-MP version 1.1 (Tracer Model 5, Massively Parallel version; Krol et al., 2005; Huijnen et al., 2010; van Noije et al., 2014; Williams et al., 2017). The model simulates the evolution of trace gases and aerosols. The model is driven by meteorological and surface fields from the ERA-Interim reanalysis produced by the European Centre for Medium-range Weather Forecasts (ECMWF) (Dee et al., 2011). A general overview of the model version applied in this work is presented by van Noije et al. (2021). The chemistry is described by mCB05, a modified version of the CB05 carbon bond mechanism (Yarwood et al., 2005) as documented in Williams et al. (2017). The developments implemented in this work comprise a treatment of reactive SOA formation and NPF in the presence of ELVOCs, which is described below.

### 2.2    M7 based aerosol model

The aerosol population and its evolution is treated with the modal two-moment model M7 (Vignati et al., 2004). The aerosol population is represented by seven log-normal modes with fixed standard deviations. Four of the seven modes represent water-soluble particles (nucleation, Aitken, accumulation and coarse) and three parallel modes represent the insoluble particles (Aitken, accumulation and coarse). The dry radii ranges for the respective modes are: Nucleation mode ($r_p < 5$ nm), Aitken mode ($5 < r_p < 50$ nm), accumulation mode ($50 < r_p < 500$ nm), coarse mode ($r_p > 500$ nm).

In this work we included secondary organic aerosols (SOA) alongside the existing species of M7, sulphate (SU), primary organic aerosol (POA), black carbon (BC), sea salt (SS) and dust (DU). Additionally, TM5 simulates the concentrations of ammonium, nitrate and methane sulfonic acid (MSA), which are represented using a bulk aerosol approach. The modelled aerosol processes include new particle formation, condensation, coagulation, wet and dry deposition together with sedimentation. Aerosol optical properties of the aerosol particles are retrieved from look-up tables, which are calculated as a function of the Mie parameter based on Mie theory (Aan de Brugh et al., 2011; Aan de Brugh, 2013). A detailed description of wet deposition can be found in de Bruine et al. (2018). For the other processes we refer the reader to van Noije et al. (2014) and van Noije et al. (2021).

### 2.3    Description of secondary organic aerosols

In earlier versions of TM5, SOA production was calculated offline (van Noije et al., 2014; Aan de Brugh, 2013; Tsigaridis et al., 2014). Using the constant mass yield reported in the Aerosol Comparisons between Observations and Models (AeroCom)

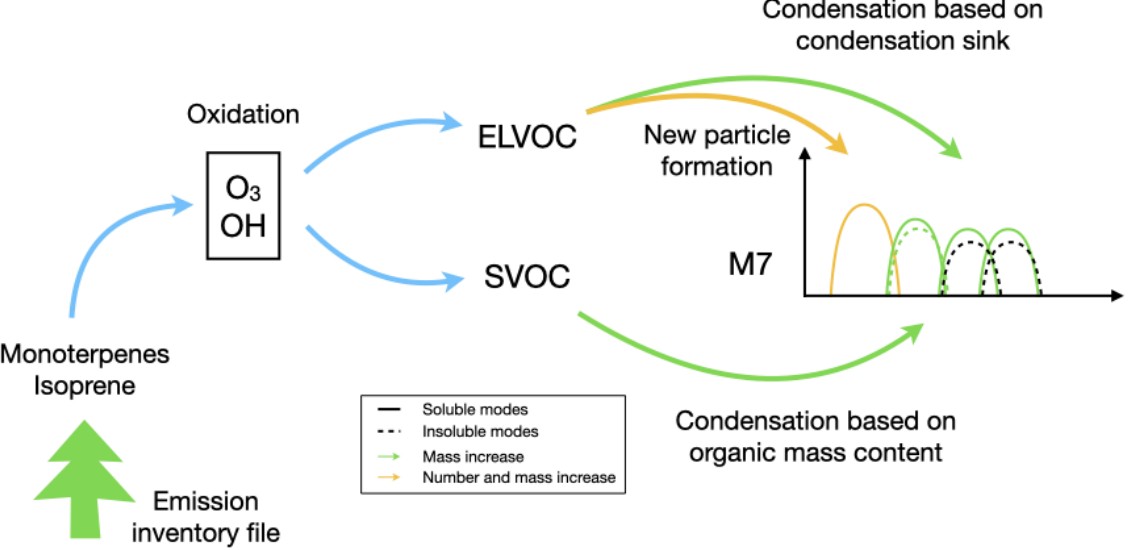

**Figure 1.** Schematic of the SOA formation scheme. Isoprene and monoterpenes are emitted from vegetation (read from emission file). Oxidation by OH and $O_3$ produces two organic surrogate species (ELVOC, SVOC). The lower volatility product ELVOC participates in new particle formation and can condense on the existing particles according to particle surface area. The semivolatile product SVOC can condense on existing particles according to their total OA mass.

project Phase I (Dentener et al., 2006) and the biogenic monoterpene emissions ($127 \mathrm{Tg(C)yr}^{-1}$) from Model of Emissions of Gases and Aerosols from Nature (MEGAN) v1 (Guenther et al., 1995), the model produced $39.9 \mathrm{Tg(SOA)yr}^{-1}$.

This freshly formed SOA mass was distributed near the surface with 80 % into heights below 30 m and 20 % in the height

ranging from 30 to 100 m (van Noije et al., 2014). The SOA mass was added to organic aerosol (OA) as additional mass into the soluble (65 % of the total) and insoluble (35 % of the total) Aitken modes with no increase in particle numbers.

The online SOA scheme described in this paper calculates the production of SOA from isoprene and monoterpenes (see Fig. 1). To track the mass of SOA in the atmosphere we have expanded M7 by including a new particulate mass tracer into all soluble modes and the insoluble Aitken mode. Despite the production of SOA being different from primary organic aerosol, we

assume otherwise the same characteristics (such as density and refractive index) for SOA as for the primary organic aerosols. The scheme is kept minimal in detail and computationally light to allow long, centennial scale integrations. In the following we detail the microphysical processes of production of gas-phase SOA precursors, their condensation and the ELVOC-induced new particle formation.

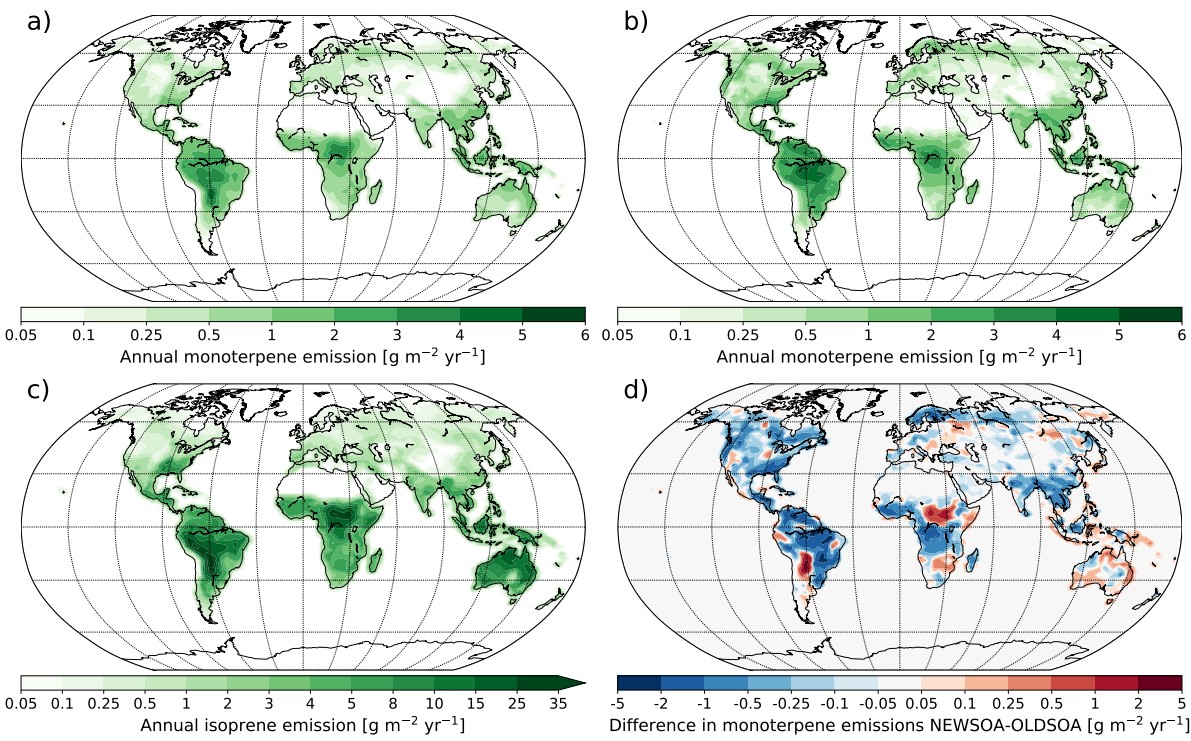

**Figure 2.** Annual emissions of monoterpene in a) new scheme, b) old scheme, c) isoprene and d) difference in monoterpene emissions (red indicates increased emissions in the new scheme).

### 2.3.1 Emissions of SOA precursors

Plants emit isoprene and monoterpenes depending on ambient conditions creating a diurnal cycle with highest emissions during daytime (Funk et al., 2003; Holzke et al., 2006). Biogenic VOC emissions calculated by the MEGANv2.1 model depend on radiation, temperature, leaf area, leaf age, and $CO_2$ (for detailed description of the model see Guenther et al., 2012). The emissions are shown in Fig. 2 (see section 2.4 for the explanation of the different simulations).

Funk et al. (2003) show that the isoprene emissions from plants follow a diurnal cycle, but due to conflicting results accurate modelling of its variation requires more research. Due to lack of a more realistic emission treatment and our dependence on emission inventories, we assume the release of isoprene during the day to vary as the cosine of the solar zenith angle with zero emissions during nighttime (Huijnen et al., 2010). The monoterpenes are emitted from storage pools mainly as a function of leaf temperature, but also other factors affect the emission such as light intensity and soil wetness (Guenther et al., 2012). Since we rely on emission inventories, the daily emissions need to be normalized which makes the use of soil wetness or temperature not feasible and therefore for monoterpene emissions we follow a similar approach as for isoprene. However, we

use a sinusoidal function which has a minimum at night and peak around noon to emulate the emission from storage pools and from photosynthesis.

In this work we employ monthly varying isoprene and monoterpene emissions (with annual emissions of 572.3 $\mathrm{Tg\,yr^{-1}}$ and 95.5 $\mathrm{Tg\,yr^{-1}}$, respectively) from an inventory derived from MEGANv2.1 (Sindelarova et al., 2014; Guenther et al., 2012, ; geographical distribution of emissions shown in Fig. 2a,c). It should be noted that the gases are emitted at ground level, i.e., in the lowest model layer. In the old scheme the calculation of SOA production is based on monoterpene emissions from an earlier MEGANv1 with an annual total of 144.2 $\mathrm{Tg\,yr^{-1}}$ (Fig. 2b). Isoprene does not contribute to the SOA formation in the old scheme. The difference to the new scheme is shown in Fig. 2d. In addition to biogenic emissions, monoterpenes and isoprene are emitted during biomass burning. These emissions are monthly emissions without diurnal variations as defined in the emission inventory (van Marle et al., 2017). At present the model does not include oceanic isoprene or monoterpene emissions, mainly due to low strength of isoprene emissions (Arnold et al., 2009) and high uncertainties of monoterpene emissions (Arnold et al., 2009; Yassaa et al., 2008).

### 2.3.2 Production of extremely low volatility and semi-volatile organic compounds

Jokinen et al. (2015) assumed the molar yield of SOA precursors from monoterpene and isoprene oxidation to be 15 % and 5 %, respectively. In the present model we separate low-volatility products into semi-volatile OC, represented by single tracer SVOC, and extremely low-volatility VOC, represented by ELVOC. Furthermore, Jokinen et al. (2015) determined experimentally the ELVOC molar yields from oxidation of monoterpenes and isoprene by ozone ($O_3$) and the hydroxyl radical (OH). The $O_3$ and OH oxidant fields are calculated online in the chemistry code.

In our implementation ELVOC and SVOC are formed in reactions of isoprene and monoterpenes with $O_3$ and OH. The reaction rate coefficients and yields for SVOC and ELVOC can be seen in Table 1. We assume two products from these reactions: extremely low-volatile organic compounds ($C_{10}H_{16}O_7$; ELVOC) and semi-volatile compounds ($C_{10}H_{16}O_6$; SVOC), which have different volatilities. In this work we assume that SVOC does not re-evaporate.

However, in contrast to Jokinen et al. (2015) we assume total production of SOA precursors from isoprene to be 1 % instead of 5 % due to very high production of SOA precursors in our inital tests with higher yields. This is in line with Kroll et al. (2005) who report 0.9–3.3 % mass yields.

Sporre et al. (2020) analyzed the behavior of the model with our NEWSOA scheme using a set of sensitivity experiments. By scaling the SVOC and ELVOC yields up and down by 50%, the resulting SOA mass was increased / decreased by almost 50% (Sporre et al., 2020, Fig. 6). Products from isoprene and monoterpenes contributed about 80% and 20% to the SOA mass (Sporre et al., 2020). The response of the number concentration and number size distributions was rather non-linear since it very much depends on the competition between new particle formation and growth of existing particles. The balance between the two processes was changed by adapting the SVOC and ELVOC yields.

**Table 1.** Rate coefficients (Atkinson et al., 2006) and molar yields used to calculate production of ELVOC and SVOC in reactions between OH and $O_3$ and monoterpene and isoprene. The yields for monoterpene reactions are taken directly from Jokinen et al. (2015) whereas the isoprene yields are lowered to produce a total yield of 1% instead of 5%, see text for further information.

| | | | Molar yield for producing | |
| Reaction | Rate coefficient [$cm^3 molecule^{-1} s^{-1}$] | Valid temperature range | ELVOC | SVOC |
| --- | --- | --- | --- | --- |
| isoprene + OH | $2.7 \times 10^{-11} \times e^{(390/T)}$ | 240 – 430 K | 0.0003 | 0.0097 |
| isoprene + $O_3$ | $1.03 \times 10^{-14} \times e^{(-1995/T)}$ | 240 – 360 K | 0.0001 | 0.0099 |
| monoterpene + OH | $1.2 \times 10^{-11} \times e^{(440/T)}$ | 290 – 430 K | 0.01 | 0.14 |
| monoterpene + $O_3$ | $6.3 \times 10^{-16} \times e^{(-580/T)}$ | 270 – 370 K | 0.05 | 0.10 |

### 2.3.3 Gas-particle partitioning of ELVOC and SVOC

Organic condensation in large scale models is generally calculated using either partitioning theory (Pankow, 1994) or kinetic condensation on the surface of aerosols (Spracklen et al., 2010). The former assumes that organic vapor molecules find equi-
175 librium instantly with the aerosol, the latter assumes that vapors are non-volatile and condensation depends on the surface area.

We apply both methods following the work of Jokinen et al. (2015). SVOCs are assumed to partition among different types of particles according to the existing total OA mass in each mode (equilibrium model) whereas ELVOCs condense according to particle surface area of each mode (kinetic approach). The equilibrium model is assumed to be irreversible, since the yields
are determined in the equilibrium state. The change in SOA mass by condensation of ELVOC and SVOC in a single mode is calculated as

$$\Delta M_{i,\text{SOA}} = \frac{CS_i}{\sum_i CS_i} \Delta M_{\text{ELVOC}} + \frac{M_{i,\text{POA+SOA}}}{\sum_i M_{i,\text{POA+SOA}}} \Delta M_{\text{SVOC}}, \tag{1}$$

where $i$ is the log-normal mode index. The first term on the right-hand side describes the condensation of ELVOC, where $CS_i$ is the condensation sink (Pirjola et al., 1999) of a single mode $i$ which is proportional to the surface area of the mode. The
185 condensation for ELVOC is applied for all soluble modes and the insoluble Aitken mode where SOA and/or POA is present. $\Delta M_{\text{ELVOC}}$ is the mass of gas-phase ELVOCs available for condensation within one timestep.

The second term on the right-hand side describes the increase in SOA mass by condensation of SVOC on particles of mode $i$, where $\Delta M_{\text{SVOC}}$ is the available gas-phase SVOC mass within one timestep, $M_{i,\text{POA+SOA}}$ is the total organic aerosol mass in mode $i$. Here we follow the original M7 modal assumption for SOA, meaning that SVOC can condense to soluble Aitken,
accumulation and coarse modes and insoluble Aitken mode. Similarly to Jokinen et al. (2015) we assume that both of these compounds have a volatility low enough that all of them will be condensed onto the existing particles within a time step of the model. Thereby we reduce the computational cost without the need to calculate the transport of gas-phase SOA precursors.

### 2.3.4 New Particle Formation

During new particle formation (NPF) low volatility gases form small molecular clusters which can transform into stable particles. This process is ubiquitous and mainly driven by sulphuric acid (Kulmala et al., 2004). Therefore, global models often parameterise NPF as function of sulphuric acid concentration (Vehkamäki et al., 2002; Sihto et al., 2006; Kulmala et al., 2006; Laakso et al., 2004). However, recent research has shown that a variety of other compounds participate in the process e.g. low-volatility VOC (Bergman et al., 2015; Riccobono et al., 2014; Paasonen et al., 2010).

In previous TM5 versions, NPF is described using the parameterisation of Vehkamäki et al. (2002) (classical nucleation theory). They calculate the nucleation rate and particle size depending on the concentration of water and sulphuric acid using a fitted formulation. In this work, we introduce an additional NPF parameterisation which takes into account gas-phase organics (Riccobono et al., 2014). This scheme formulates the formation of particles of 1.7 nm in diameter as a function of the concentrations of sulphuric acid ($H_2SO_4$) and oxidised biogenic compounds (BioOxOrg), which is based on measurements done in the CLOUD chamber (Cosmics Leaving OUtside Droplets) at CERN (Kirkby et al., 2011). In our implementation we represent the BioOxOrg as ELVOC following Eq. 2 in Riccobono et al. (2014) with p=2 and q=1, as

$$J_{\mathrm{RICCO}} = K_m [\mathrm{H_2SO_4}]^2 [\mathrm{ELVOC}], \tag{2}$$

where $K_m = 3.27 \times 10^{-21}$ cm$^6$s$^{-1}$ is an empirical factor and $[\mathrm{H_2SO_4}]$ and $[\mathrm{ELVOC}]$ are the gas-phase concentrations of sulfuric acid and ELVOCs, respectively. Whereas BioOxOrg represents the products from oxidation of monoterpenes by OH, our ELVOC includes oxidation products from isoprene and monoterpenes.

The new scheme combines two different parameterisations to calculate the NPF: the binary homogeneous water - sulphuric acid nucleation (Vehkamäki et al., 2002, referred to as BHN in the following) and the semi-empirical parameterisation by Riccobono et al. (2014) (referred to as RICCO in the following).

### 2.3.5 Parameterisation of particle growth to 5 nm

The growth of small particles in M7 is hindered by the modal structure of the model (Korhola et al., 2014). Therefore, we calculate the parameterised NPF rate for particles of 5 nm in diameter using the formulation of Kerminen and Kulmala (2002, KK in the following) in four phases (see schematic in Fig. 3). First, we calculate the BHN rate ($J_{\mathrm{BHN}}$) and associated diameter of the formed particles. Second, to combine the formation rate of the BHN and the RICCO schemes the formation rate of the BHN scheme is required for particles of the same size (i.e. 1.7 nm). KK is then used to calculate the growth of particles by ELVOC and sulfuric acid vapors from BHN formation size to 1.7 nm diameter particles. In the third step, the formation of 1.7 nm particles according to RICCO is calculated and added to the one from BHN. Finally, we use KK again to calculate the growth from 1.7 nm to 5 nm due to condensation of ELVOC and sulfuric acid. A more detailed description can be found in Appendix A.

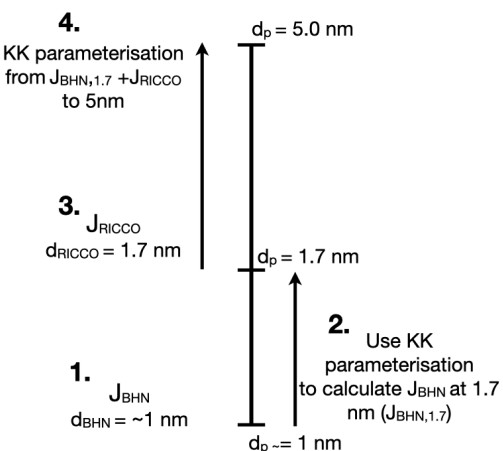

**Figure 3.** Schematic of the four phases of the parameterised calculation of new particle formation rates of 5 nm particles. Phase 1: Calculation of nucleation rate of the BHN scheme ($J_{BHN}$) which is the formation rate by binary homogeneous water-sulfuric acid nucleation, Phase 2: Calculation of the parameterised growth by Kerminen and Kulmala (2002) to 1.7 nm, Phase 3: Calculation of semi-empirical particle formation rate $J_{RICCO}$ considering ELVOC, Phase 4: Calculation of the parameterised growth by (Kerminen and Kulmala, 2002) from 1.7 nm to 5 nm.

## 2.4 Simulations

We have run three simulations with TM5 to evaluate the impact of the new SOA scheme on the organic mass and particle
number concentrations. The first simulation, called OLDSOA, is done with the old formulation where all SOA mass is added
to the primary OA in Aitken mode as explained in the beginning of section 2.3. This simulation uses prescribed SOA pro-
duction based on monoterpene emissions from the older MEGANv1 (Guenther et al., 1995).Isoprene does not contribute to
the SOA formation in the old scheme. The second simulation, called NEWSOA, utilises the SOA description given in this
section.Biogenic monoterpene and isoprene emissions for SOA production in NEWSOA use MEGANv2.1 (Sindelarova et al.,
2014; Guenther et al., 2012). For all other biogenic emissions we use MEGANv2.1 (Sindelarova et al., 2014; Guenther et al.,
2012) in all three simulations. The third simulation OLDSOA-MEGAN2 is a sensitivity simulation to evaluate the effect of
using MEGANv2.1 monoterpene emission but still applying the SOA yield used in the old scheme.

    Oceanic DMS emissions are calculated online using dimethylsulfide (DMS) concentrations from Lana et al. (2011) which
uses the exchange velocity from Wanninkhof (2014). Emissions of mineral dust are based on Tegen et al. (2002). A detailed
description of DMS and dust emissions will be described in van Noije et al. (2021). Sea salt emissions are calculated based on
the parameterisation of Gong (2003) with wind speed dependence from Salisbury et al. (2013). These emissions are described
in more detail by van Noije et al. (2021). Other natural emissions are prescribed as in van Noije et al. (2014). For the emissions

**Table 2.** Emissions of precursors gases and particulate phase compounds in the simulations NEWSOA and OLDSOA. Numbers are indicated in Tg per year for emissions.

| | Emitted/produced mass | References |
|---|---|---|
| BC | 9.68 | Hoesly et al. (2018); van Marle et al. (2017) |
| OA | 55.43 | Hoesly et al. (2018); van Marle et al. (2017) |
| SOA Production (for more details see Table 3) | | |
| NEWSOA | 52.48 | precursor emissions from Sindelarova et al. (2014) using new scheme |
| emission of monoterpenes | 95.5 | |
| emission of isoprene | 572.1 | |
| OLDSOA | 39.90 | precursor emissions from Guenther et al. (1995) using old scheme |
| emission of monoterpenes | 127 | |
| OLDSOA-MEGAN2 | 30.86 | precursor emissions from Sindelarova et al. (2014) using old scheme |
| emission of monoterpenes | 95.5 | |
| SO4 emission | 4.72 | Hoesly et al. (2018); van Marle et al. (2017) |
| SO2 emission | 122.8 | Hoesly et al. (2018); van Marle et al. (2017) |
| DMS emission | 51.1 | Lana et al. (2011); Wanninkhof (2014); van Noije et al. (2021) |
| Sea salt | 5533.93 | Gong (2003); Salisbury et al. (2013); van Noije et al. (2021) |
| Dust | 1124.09 | Tegen et al. (2002); van Noije et al. (submitted to Geoscientific Model Dev |

of gases and particulate matter from anthropogenic sources and biomass burning we use the CMIP6 input4MIPs inventory (Hoesly et al., 2018; van Marle et al., 2017).

For the simulations we have used a horizontal resolution of 3° longitude by 2° latitude and 34 hybrid-sigma levels. The model is driven by the ERA-Interim reanalysis produced by the ECMWF (Dee et al., 2011). All three simulations are run for the year 2010 with a 11 month spin-up period.

## 2.5 Observational data used in model evaluation

In order to evaluate the impact of the new SOA scheme on the simulated aerosol properties, the model is compared against observations of organic mass concentrations, number concentrations and AOD. The observational data from surface measurements, in-situ remote sensing and satellite retrievals are described below.

### 2.5.1 Organic mass concentrations at the surface

We evaluate the model performance of simulating OA concentrations at the surface by comparing to two freely available observational networks, the United States' Interagency Monitoring of Protected Visual Environments (IMPROVE; http://vista.cira.colostate.edu/improve/ last access 11.1.2018; Malm et al., 1994) and European monitoring and evaluation project (EMEP; http://www.emep.int; last access 27.7.2017; Tørseth et al., 2012). For the IMPROVE network we use the organic mass in PM2.5 particles from 175 stations (see Table S1 for a list of stations) and for EMEP we use PM2.5 or PM10 depending on the station for 15 stations (see Table S2 for a list of stations). The sum of simulated primary and secondary organic aerosol concentrations in the lowest model layer has been collocated with the location, and time of the observations. However, Schutgens et al. (2016a) have shown that comparisons of in-situ measurements of aerosol mass concentrations to simulated concentrations will show significant errors. However, here we report aggregated monthly and yearly means of the observed and simulated values. In the figures we use standard error of the mean for modelled and observational data.

Both EMEP and IMPROVE networks measure particulate organic carbon (OC) instead of total organic mass in the particles. Therefore, the carbon content is usually converted to organic mass with a constant factor. For the whole IMPROVE network the suggested ratio between carbon and particulate organic matter in PM2.5 particles is 1.8 (Pitchford et al., 2007, http://vista.cira.colostate.edu/Improve/the-improve-algorithm/), which is used in our analysis also. The conversion factor from OC to OA at the European sites is commonly assumed to be 1.4 (Putaud et al., 2004; Sillanpää et al., 2005). However, since Yttri et al. (2007) show that usually the ratio between OC and OA should be higher than that, we follow the IMPROVE network implementation and assume a factor of 1.8 also for the EMEP stations. It has to be noted that in the primary emitted carbon for POA is converted to total mass with constant factor of 1.6 (van Noije et al., 2021).

### 2.5.2 Number concentrations onat the surface

Aerosol number concentrations from several observation campaigns are hosted at EBAS webservice (ebas.nilu.no; Tørseth et al., 2012). The stations there provide condensation particle counter (CPC) observations around the globe, but the coverage is very sparse. In this dataset, USA and Europe are overrepresented while most of Asia is lacking observations. Nevertheless, we have collocated the simulated concentrations of particles with diameter larger than 10 nm in time and space to 27 stations (see Table S3 for a list of stations), where number concentration data was available for 2010 to compare the annual mean from NEWSOA and OLDSOA simulations to the observations (Fig. 8).

### 2.5.3 Remote sensing data

The comparison to satellite retrievals on aerosol optical depth (AOD) provides the opportunity to evaluate the new SOA description globally. Here we use the AOD products from two different satellite instruments: the MODerate resolution Imaging Spectroradiometer (MODIS) retrieval and the Advanced Along Track Scanning Radiometer (AATSR) retrieval.

MODIS is located aboard two satellites Aqua and Terra providing good coverage in the morning and afternoon (King et al., 1999). We use here the combined product from Deep Blue and Dark Target algorithms in MODIS C6 (Sayer et al., 2014) data

from both satellites. According to recommendation by Schutgens et al. (2016b) the modelled AOD data is collocated in space
and time to the satellite retrieval using CIS tools (www.CIStools.net; Watson-Parris et al., 2016).

The AATSR onboard ESA's Envisat satellite provides 35 day periodic coverage for the globe. Here we use the aerosol optical
depth retrieval by the Swansea University (Popp et al., 2016), which was chosen due to best ranking over land in de Leeuw
et al. (2018).

The sun-photometer network AERosol RObotic NETwork (AERONET; Holben et al., 1998) provides global coverage of in-
situ measurements of AOD, although in many areas the network is rather sparse. Nonetheless, these observations are considered
as a ground truth in aerosol optical depth (AOD) retrievals. Since AOD at 550 nm is not available from many AERONET sites,
their instantaneous aerosol optical depths at 550 nm ($AOD_{550}$) are derived using Ångström power law, Ångström exponent
(440-675 nm; $\alpha$) and level 2 AERONET AOD at 500 nm ($AOD_{500}$):

$$AOD_{550} = AOD_{500} \left( \frac{\lambda_{550}}{\lambda_{500}} \right)^{-\alpha} \tag{3}$$

We have collocated the simulated hourly AOD at 550nm with the AERONET AOD in space and time to provide us with the
best possible evaluation of the model performance. In our comparison we have included all 299 stations which provide data for
the year 2010. After the collocation we show the aggregated monthly and yearly mean data. This spatio-temporal collocation
was done to limit the errors using hourly data when possible. However, as Schutgens et al. (2016b, a, 2017) state some error
will remain.

## 3 Results

### 3.1 Changes in the global SOA budget

Table 3 shows the annual SOA budget for OLDSOA and NEWSOA simulations together with some literature values. The
annual production of SOA in NEWSOA was increased by 35.0 % to 52.5 Tg(SOA) yr$^{-1}$ from 39.9 Tg(SOA) yr$^{-1}$ in OLD-
SOA. When using the MEGANv2.1 monoterpene emissions in OLDSOA the annual mean production is reduced to 30.9
Tg(SOA) yr$^{-1}$. The new annual SOA production is within the range of other global estimates. It falls within the 12–70
Tg(SOA) yr$^{-1}$ estimated by Kanakidou et al. (2005) and the $25 - 210$ Tg(C) yr$^{-1}$ (about 50–420 Tg(SOA) yr$^{-1}$ assuming
double mass for total SOA) estimated by Hallquist et al. (2009), while it is below the estimate of $140 - 910$ Tg(C) yr$^{-1}$ (about
280–1820 Tg(SOA) yr$^{-1}$ assuming double mass for total SOA) estimated by Goldstein and Galbally (2007). Furthermore,
our SOA formation in NEWSOA is close to the lower limit of Spracklen et al. (2011) range 50–380 Tg(SOA) yr$^{-1}$ and well
below their best estimate of 140 Tg(SOA) yr$^{-1}$. However, their work includes anthropogenic SOA, which is not included in
our model. Stadtler et al. (2018) used a complex isoprene chemistry to produce 138.5 Tg(SOA) yr$^{-1}$ SOA, which is much
higher than our total from monoterpenes and isoprene. Furthermore, NEWSOA is in line with the the mean SOA production
of 59 Tg yr$^{-1}$ of the 12 models with online SOA production used in an AeroCom multi-model evaluation of OA by Tsigaridis
et al. (2014).

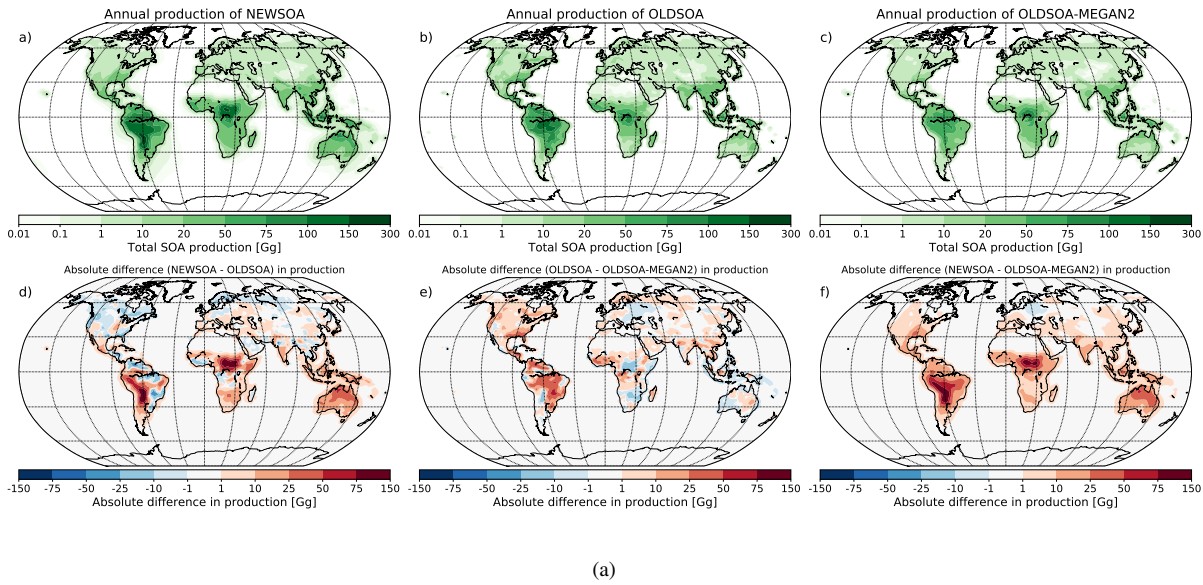

**Figure 4.** Annual mean production of SOA in NEWSOA (a) , OLDSOA (b) and OLDSOA-MEGAN2 (c) and absolute difference in the global annual production of SOA between simulations NEWSOA and OLDSOA(d), OLDSOA-MEGAN2 and OLDSOA (e) and NEWSOA and OLDSOA-MEGAN2. In the difference plots red areas indicate increased production in NEWSOA compared to OLDSOA.

The total SOA production here is roughly double to that predicted by Jokinen et al. (2015). In addition, the shares of ELVOC and SVOC as well as the shares of precursors are quite different. In their model the ELVOC production is only $0.9 \, \mathrm{Tg \, yr^{-1}}$, while in our implementation it is $7.3 \, \mathrm{Tg \, yr^{-1}}$. Thereby, also the ELVOC fraction is higher in our case, with 14 % of total production being ELVOC compared to 3.2 % by Jokinen et al. (2015). For the precursors they show 12 % of SOA originating from isoprene while in our NEWSOA simulation it is 47 %. Therefore, even though both models employ very similar SOA schemes, different emissions and chemistry cause distinct differences in the global annual mean production of SOA.

When comparing the regional differences in SOA production between our two main simulations NEWSOA and OLDSOA in Figs. 4a and 4b (seasonal production can be found in Figs S1 and S2), the emission regions remain mostly the same. Strongest production is found in the rain forests of South America and Africa and the lowest production over the deserts, most notably in the Sahara and Gobi deserts. However, there are some changes in the main production regions in South America and Africa. Additionally, in Australia the production increases strongly in NEWSOA compared to OLDSOA resulting from inclusion of isoprene in NEWSOA (Fig. 4d). Another interesting change is in the northern high latitudes, where the SOA production in NEWSOA is lower than in OLDSOA throughout the year (Figs. S3 and S4), with clearly lower production in boreal regions in Scandinavia, northern Canada and northern Russia. A comparison of the main simulations to the sensitivity simulation OLDSOA-MEGAN2 (Figs. 4d,e) shows that OLDSOA has higher production around the globe except at some locations. Especially the Amazon region shows higher production in OLDSOA. In Eastern Europe the production in OLDSOA-

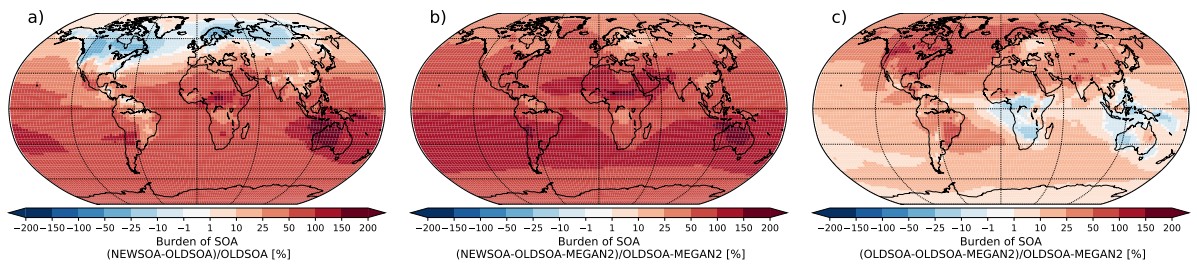

(a)

**Figure 5.** Fractional change in atmospheric burden of secondary organic aerosol between simulations NEWSOA and OLDSOA (a) OLDSOA and OLDSOA-MEGAN2 (b), and NEWSOA and OLDSOA-MEGAN2 (c).

MEGAN2 is increased compared to both main simulations. At northern high latitudes the production has similar (or slightly lower) magnitude for the NEWSOA and OLDSOA-MEGAN2 simulations, but elsewhere NEWSOA gives a higher production.

Mainly the differences are caused by using SOA production calculated from older Guenther et al. (1995) terpene emissions using MEGANv1 in the OLDSOA simulation, while NEWSOA uses MEGANv2.1 (see also Sect 2.3.1; Guenther et al., 2012; Sindelarova et al., 2014) isoprene and monoterpene emissions. In addition, some difference can be caused due to the use of oxidation by OH and $O_3$ to calculate the production of SOA from the precursor concentrations in the atmosphere. Fig. 4 shows that the SOA production in NEWSOA is increased over most of the globe compared to OLDSOA-MEGAN2, with the exception of northern high latitudes. Note that there the fixed SOA aging produces similar amounts as the online calculation of the monoterpene oxidation. Furthermore, in the boreal region the isoprene oxidation contributes mostly only about 10-20 % of the total SOA production.

### 3.1.1 Atmospheric SOA budget

Figure 5 shows the change in atmospheric burden of SOA between NEWSOA and OLDSOA (a) (seasonal mean atmospheric burden shown in Fig. S5), between NEWSOA and OLDSOA-MEGAN2 (b), and between OLDSOA and OLDSOA-MEGAN2 (c). The annual mean SOA burden is increased by 60.0 % from 0.85 Tg in OLDSOA to 1.31 Tg in NEWSOA. The burden in NEWSOA is somewhat higher than the average AeroCom model ensemble (Tsigaridis et al., 2014) and higher than the estimate of 0.85 Tg by Henze et al. (2008) but lower than the best estimate of 1.84 Tg by Spracklen et al. (2011). The sensitivity simulation OLDSOA-MEGAN2 shows a decrease in annual burden to 0.70 Tg.

Although the global annual mean burden increases in NEWSOA compared to OLDSOA, in the northern high latitudes in Canada, Scandinavia and Russia it is reduced by 10 %- 90 %, which is expected since the production in the northern high latitudes has decreased significantly as noted above. Even though the production in these regions in OLDSOA-MEGAN2 and

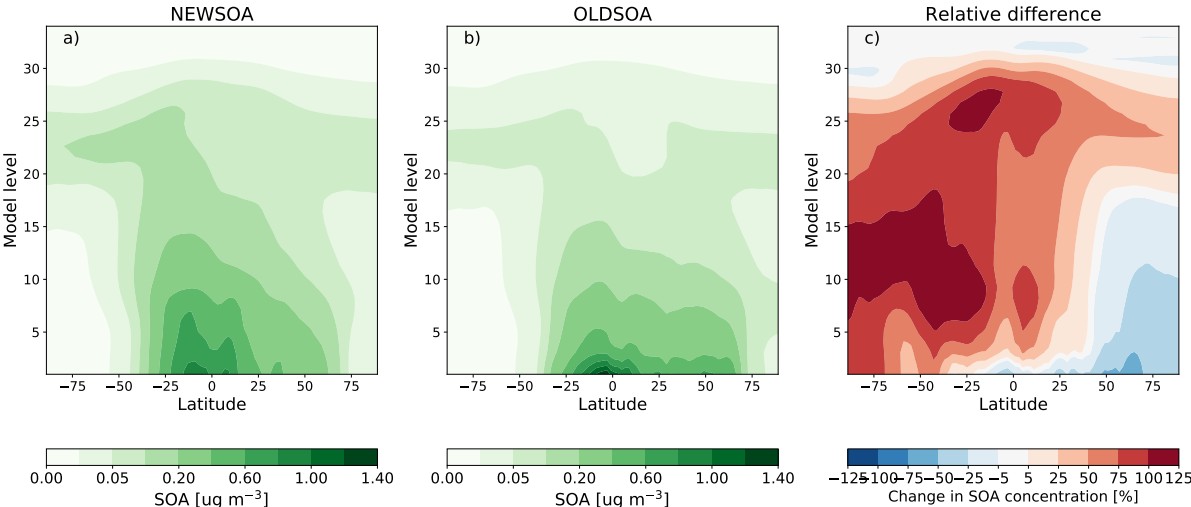

**Figure 6.** Zonal annual mean concentration of SOA in simulations NEWSOA (a) and OLDSOA (b), and the relative change from OLDSOA to NEWSOA (c).

NEWSOA are similar (see 4f) the atmospheric burden is similar only in Western Russia and the Baltic region (see Fig. 5b) highlighting the difference in distribution of organic mass onto particles. In the Southern hemisphere (SH) the increase in SOA burden is more than 50 % almost everywhere, which is inline with the strong increase in production in the SH. A lower increase of 25-50 % can be seen in sub-Saharan Africa and eastern Brazil. In contrast, over Australia a strong increase in SOA burden
can be observed (more than 100 %), which is expected due to strong isoprene emissions (Guenther et al., 2006) and subsequent SOA formation from isoprene, which is not accounted for in OLDSOA.

The zonal annual mean concentrations of the SOA in simulations OLDSOA and NEWSOA and their difference are shown in Figure 6. On average the concentrations of SOA are higher near the Earth's surface in OLDSOA than in NEWSOA, which is expected, because in OLDSOA all of the produced SOA mass is distributed into the lowest 100 m of the boundary layer. In
the Southern Hemisphere the concentrations aloft are as much as 100 % higher in NEWSOA compared to OLDSOA as can be seen in Fig. 6c. Higher SOA concentrations aloft in NEWSOA are expected due to the SOA production occurring throughout the atmosphere from oxidation products of monoterpenes and isoperene instead of emitting it into the boundary layer below 100 m as in OLDSOA. In contrast, the figure shows clearly that the concentrations are lower in NEWSOA in the northern high latitudes (50-75 % lower) due to the lower production of SOA in the northern Hemisphere in NEWSOA. This results mainly
from spatially different emissions of monoterpenes in MEGANv1 compared to MEGANv2.1 as stated earlier (see Fig. 2).

The removal mechanism of SOA is almost exclusively wet deposition (98.9 %). It has increased by 12.8 $\mathrm{Tg\,yr^{-1}}$ (38.5 %) in NEWSOA compared to the earlier scheme mainly due to higher production in NEWSOA. Although a high fraction of wet deposition is expected (Flossmann and Pruppacher, 1988), the removal fraction by wet deposition is clearly higher than the contribution of 85 % observed in the AeroCom multi-model ensemble (Tsigaridis et al., 2014). However, in Stadtler

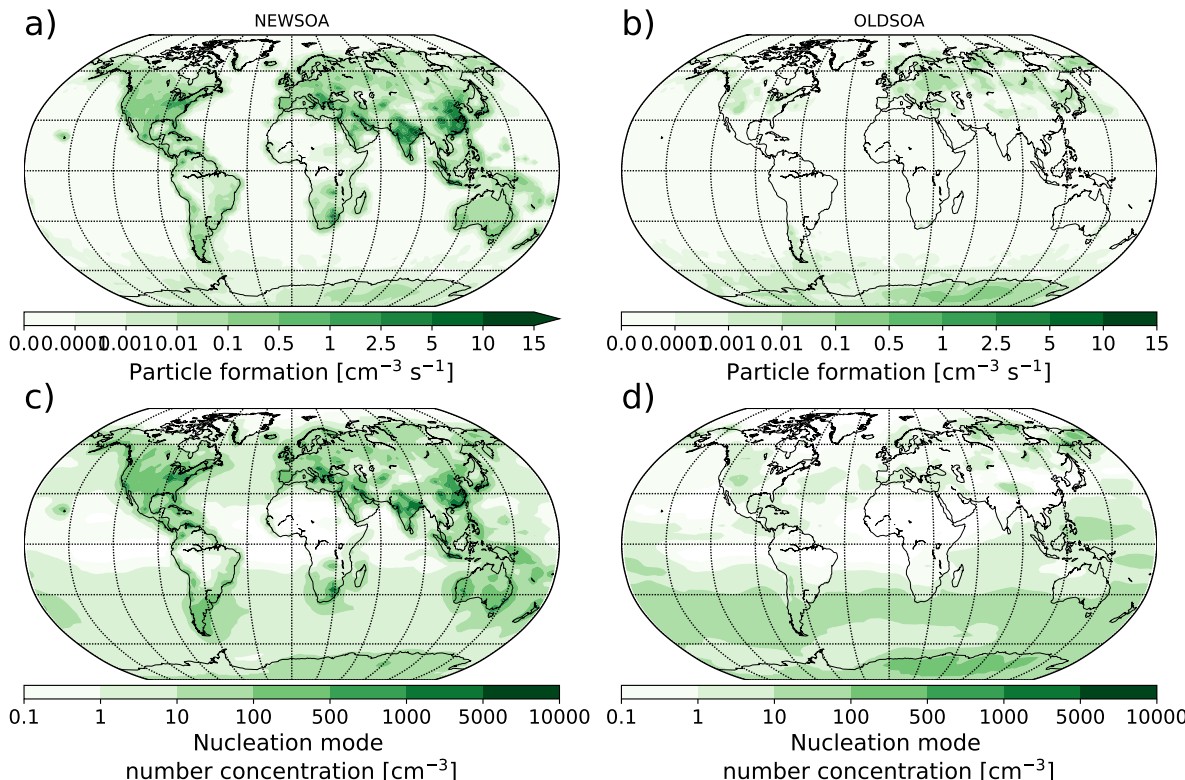

**Figure 7.** Modelled annual mean new particle formation rates at the surface are shown in the top row for the runs NEWSOA (a) and OLDSOA (b). The bottom row shows the surface layer concentration of particles in the nucleation mode in runs NEWSOA (c) and OLDSOA (d).

et al. (2018) the ECHAM-HAMMOZ model using online formation of SOA from isoprene shows high wet deposition (96.8%) similarly to our results.

The lifetime of SOA in Table 3 is calculated as a ratio between burden of SOA over its annual mean removal. It has increased in NEWSOA by about one day from 7.8 to 9.1 days. In NEWSOA the SOA mass is distributed into the aerosol size distribution more evenly than in OLDSOA. This means that mass in the accumulation mode is decreased while the mass in the Aitken mode

is increased, which could suggest a lower lifetime (see Schutgens and Stier (2014) for mode-wise lifetimes in M7). However, in NEWSOA the SOA production aloft is clearly higher meaning that the deposition takes more time leading to a longer lifetime.

Although wet deposition, burden and lifetime fall within the multi-model ranges of Tsigaridis et al. (2014), the dry deposition is clearly lower than either their multi-model mean or multi-model range. Therefore, it would be worthwhile to examine the wet and dry deposition processes of SOA in TM5 in more detail in a dedicated study.

## 3.2 Changes in new particle formation (NPF)

As explained in Sect. 2.3.4, in the previous version of TM5 the NPF is calculated using the parameterisation of Vehkamäki et al. (2002), which is not able to produce the observed particle number concentrations in the boundary layer (Spracklen et al., 2006; Makkonen et al., 2009). Here we show how the particle formation and number concentrations change using the additional particle formation from organically enhanced new particle formation by Riccobono et al. (2014).

Figure 7a,b shows the modelled annual mean new particle formation rate at the surface for NEWSOA and OLDSOA. The annual mean formation rate at the surface is $0.021 \, \mathrm{cm^{-3} \, s^{-1}}$ and $0.00021 \, \mathrm{cm^{-3} \, s^{-1}}$ for NEWSOA and OLDSOA, respectively. The binary homogeneous nucleation produces very few particles near the surface except in Siberia and Antarctica. Since the RICCO NPF parameterisation depends on the ELVOC and sulfuric acid concentrations, formation rates over the oceans are negligible (below $1 \times 10^{-4} \, \mathrm{cm^{-3} \, s^{-1}}$). Over land areas of highest formation rates mostly coincide with locations of high ELVOC production. However, the formation rates in the Amazon and sub-Saharan Africa are very low although ELVOC production rate is high. These locations have low sulfuric acid concentrations, which in turn limits the new particle formation here. Contrarily, the Arabian peninsula shows high formation rates due to high sulfuric acid concentrations although ELVOC concentrations are relatively low.

Figure 7c,d show the concentration of particles in nucleation mode (below 10 nm in diameter) in the two simulations NEWSOA and OLDSOA. The highest particle concentrations are seen in regions where the organically enhanced NPF leads to high formation rates of new particles. However, the NPF rate is very low over oceans and Antarctica; regardless of that particles smaller than 10 nm in diameter are present over these areas. Here the particles are entrained from the free and upper troposphere, where the BHN is more efficient at producing particles. This behaviour is in line with earlier research (Spracklen et al., 2005; Korhonen et al., 2008; Merikanto et al., 2009). Furthermore, OLDSOA shows higher concentrations (and NPF) over Antarctica. The reason for this is the different handling of nucleation mode particles in the two main simulations; in NEWSOA the particle growth to 5nm in diameter is parameterized while in OLDSOA no growth of particles is assumed (see Sect. 2.3.5 for details).

## 3.3 Particle number concentrations

Figure 8 shows the annual means from NEWSOA and OLDSOA simulations compared to the observed particle number concentrations at the stations listed in Table S3. In general the concentrations are higher in NEWSOA with a mean increase of 14 % across the stations, although the absolute differences are small. The absolute annual mean across stations ($1460 \, \mathrm{cm^{-3}}$) is underestimated with OLDSOA ($1350 \, \mathrm{cm^{-3}}$) and overestimated in NEWSOA ($1540 \, \mathrm{cm^{-3}}$), meaning that the low bias in OLDSOA with normalized mean bias (NMB) of -7.2 % has changed to a high bias in NEWSOA with NMB of 5.4 %. In NEWSOA the fraction of concentrations within a factor 2 of the observed values is 63 % while in OLDSOA this is only 44 %. In addition, the correlation coefficient increases from 0.50 in OLDSOA to 0.57 in NEWSOA, showing some improvement from the old scheme. Summarising, the collocated annual means show limited improvement.

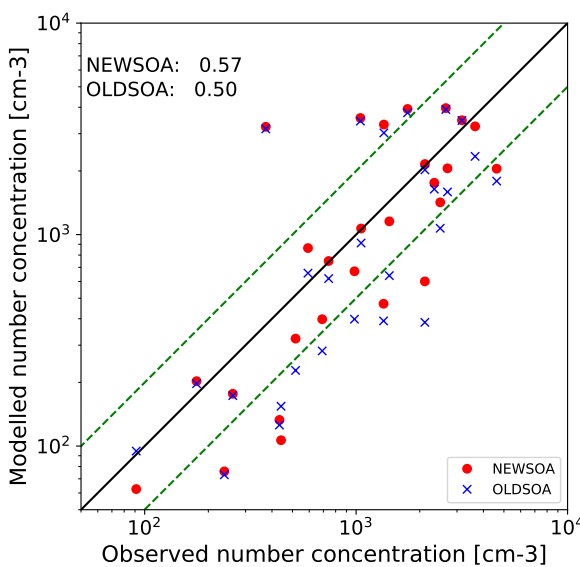

**Figure 8.** Scatter plot of collocated annual mean number concentrations at the stations in the year 2010. Red indicates NEWSOA and blue OLDSOA. The black solid line shows the 1:1 line and the green dashed lines a deviation of a factor of 2. Numbers in the corner indicate the correlations of model runs to observations.

In general the updated SOA and new NPF parameterisation caused an increase in particle number concentrations, but at two Antarctic stations the concentrations decreased. The main reason is the change in the NPF parameterisation where in OLDSOA the NPF production is calculated directly by BHN nucleation and in NEWSOA the early growth to 5 nm diameter particles is parameterised (see Sect. 2.3.5), reducing the number of particles in the nucleation mode.

### 3.3.1 Particle size distributions at selected sites

Considering the small amount of continuous long-term comprehensive observations of the aerosol-chemical system, it remains extremely difficult to constrain how distinct aerosol dynamical processes perturb the particle size distribution, and how those processes are influenced by regional and large scale physical and chemical conditions. From a modelling perspective, a large set of simulations with varying parameter values (Perturbed Parameter Ensemble, PPE) would allow to assess the sensitivity of the simulated size distributions to underlying parameter uncertainties, although limited to a single model framework (e.g. Sengupta et al., 2020). Such experiment is outside the scope of this paper, and we limit ourselves to presenting selected cases which highlight the size distribution properties simulated by the improved TM5 model in simulation NEWSOA.

Figures 9a and 9b show two separate growth events from Birkenes, Norway. Both simulated cases show a clear nucleation event with continued growth in Aitken mode. NEWSOA likely underestimates formation rates during the event on 24.4.2010 (Fig. 9b), since observed N20 (particles with particle diameter $d_p > 20$ nm) reaches even 5000 $\mathrm{cm}^{-3}$ which is higher than

total condensation nucleii (CN) peak in the NEWSOA simulation during that day. Nevertheless, NEWSOA simulates aerosol growth from nucleation to Aitken and accumulation modes, resulting in cloud concensation nucleii (CCN) peak on 25th of April. During 24.–29.9.2010 (Fig. 9a) NEWSOA simulates the evolution of both CN and CCN except for peak concentrations.

In Vavihill, during the first week of July (Fig 9c), NEWSOA successfully predicts nucleation events but seems to underestimate nucleation or subsequent growth to the Aitken mode, or both. During 2nd of July, nucleation and resulting N3 (particles with $d_p > 3$ nm) concentrations match the observations and clear growth to Aitken mode is simulated. Nevertheless, simulated growth to accumulation mode remains slower and less efficient, rendering N70 (particles with $d_p > 70$ nm) concentrations significantly lower than observed values.

During a few days in mid-June, NEWSOA simulates nucleation and growth events in Harwell (Fig. 9d). While the observed Aitken-mode growth is not visible in the simulated distribution, N20 peak concentrations are well simulated, although peak times deviate from the observed one. In Waldhof (Fig. 9e), aerosol formation rates are clearly underestimated even though the observed size distribution starts at 20 nm. Only a minor contribution from nucleation to N20 and N70 is simulated, and the increased trend in CCN is due to a shift in airmass trajectories. Both simulations and observations in Mace Head show a distinct aerosol size distribution for marine and continental airmasses. Fig. 9f shows an example of a simulated nucleation event (15th October 2010) with visible growth in the Aitken mode. The sustained growth leads to an increase in N70 during the course of 12 hours.

To summarize, TM5 with modal microphysics including improved nucleation and SOA mechanisms is able to capture nucleation and growth events but effective formation of CCN-sized particles from new particle formation events might remain limited due to numerical challenges e.g. errors from operator splitting, unrealistic size distributions due to mode merging (Whitby et al., 2002; Wan et al., 2013; Korhola et al., 2014), underestimated nucleation rates, and underestimated concentrations of vapours available for sub-CCN growth. However, it should be recognized that we do not expect a one-to-one match between a coarse-grid global model and local aerosol observations (Schutgens et al., 2016a).

## 3.4 Organic mass concentrations at the surface

In this section we compare the simulated surface concentrations of organic mass in PM2.5 and PM10 particles to EMEP and IMPROVE network observations which are described in Section 2.5.

### 3.4.1 Evaluation at IMPROVE stations

Figure 10 shows the monthly mean organic mass in PM2.5 across the IMPROVE stations in the NEWSOA (a), OLDSOA (b) , OLDSOA-MEGAN2 (c) simulations. OLDSOA and OLDSOA-MEGAN2 shows an increase in concentrations during summer, which is not present in either the observations or NEWSOA. In the sensitivity simulation OLDSOA-MEGAN2 the peak has lower magnitude, which is expected due to lower precursor emissions. This peak is caused by the addition of all SOA mass in the Aitken mode without any increase in particle number. Therefore the particles in Aitken mode grow quickly to accumulation mode, which has a lower deposition rate (Seinfeld and Pandis, 2006). In NEWSOA, however, the SOA mass is condensed more realistically across the particle size distribution using either kinetic or thermodynamic assumptions (see Section 2.3.3).

Therefore, in NEWSOA we see behaviour similar to the observations. This improvement in the representation of organic mass in NEWSOA reduces the deviation between observations and model substantially (NMB from 184 % in OLDSOA to -18 % in NEWSOA, see Table 4) and results in a correlation coefficient R=0.59. The strong overestimation (NMB=184 %) of the OLDSOA run, also seen in the scatter plot (Fig. 10d), is removed and the correlation with observations is improved notably. OLDSOA-MEGAN2 shows a similar correlation as NEWSOA but the the annual concentrations still show positive NMB of

57%. The change in emissions does not correct the seasonal cycle illustrating that a process-based SOA scheme is needed.

The new formulation is now underestimating the OA concentration slightly. However, it does not include any anthropogenic SOA nor the production from oxidation by $NO_3$. In addition, the conversion factor for primary emitted carbon to organic mass (1.6) in the model is lower than the conversion factor used by the IMPROVE network (1.8), which has a contribution to the underestimation.

### 3.4.2 Evaluation at EMEP stations

The simulated monthly mean surface concentrations of the OA mass averaged over the EMEP stations are shown and compared with the measurements in the two top rows of Fig. 11 for NEWSOA (a,d) and OLDSOA (b,e) and OLDSOA-MEGAN2 (c,f) with PM2.5 on the top row and PM10 on the second row. The annual mean concentrations are compared in the bottom row of Fig. 11 for NEWSOA (g), OLDSOA (h) and OLDSOA-MEGAN2 (i).

The seasonal cycle at the EMEP stations does not show such a dramatic change from OLDSOA to NEWSOA as for the IMPROVE stations for either PM2.5 or PM10. Overall, the rather realistic concentrations for PM2.5 in OLDSOA with NMB=-7 % are reduced to a low bias with NMB=-26 % in NEWSOA, and for PM10 concentrations in NEWSOA show similarly a reduction in NMB from -3 % to -17 %. Furthermore, OLDSOA-MEGAN2 shows similar pattern as NEWSOA for PM2.5 an NMB of -23%, indicating that a large fraction of the change is due to changes in the emissions of monoterpenes. In NEWSOA

and OLDSOA simulations the seasonal cycle is reasonably well captured for PM10, and the winter concentrations are realistic for both PM2.5 and PM10. However, in NEWSOA the summer concentrations are on the low side for PM2.5 and PM10. Furthermore, despite the stronger low bias for PM10 in NEWSOA, the summer concentrations in NEWSOA show similar correlation (difference of 0.03) as in OLDSOA. The correlation of modelled PM2.5 shows a strong reduction (down by 0.16) due lower summertime concentrations in NEWSOA than those in OLDSOA.

For the annual mean the situation is mostly similar with NEWSOA underestimating the surface concentrations of PM2.5 and PM10 (NMB=-28 % and NMB=-18 %, respectively) more strongly than OLDSOA (NMB=-10 % and NMB=-6 %, respectively). The root mean square error is slightly increased in the NEWSOA simulation (from 1.72 to 1.86) for PM2.5 while it has decreased for PM10 (from 2.14 to 2.00).The sensitivity simulation OLDSOA-MEGAN2 has values in between, but closer to NEWSOA. This means that much of the change is due to lower emissions in Europe than when using MEGANv1

emissions. However, as for IMPROVE it is expected that the total OA is underestimated, since the model doesn't account for antropogenic SOA and other biogenic SOA sources (e.g. VOC oxidation by $NO_3$). In addition, the conversion factor of POA carbon to total organic mass (1.6) in the model is lower than than the one used for the station data (1.8), which has a small contribution to the underestimation. Furthermore, the good local agreement in OLDSOA with prescribed SOA formation can

be accidental, while online oxidation in NEWSOA takes into account the oxidant levels. Therefore the missing sources may lead to underestimations in certain areas.

### 3.4.3 Summary of surface organic mass concentrations

In NEWSOA the OA burden has been reduced in the Northern Hemisphere and increased in the Southern Hemisphere as shown in Fig. 12a. At the surface OA concentration (Fig. 12b) is mostly increased in the Southern Hemisphere. However, in South America and Africa the surface concentrations, even with roughly similar production of SOA (see Fig. 4), are lower due to the NEWSOA production occurring more aloft while OLDSOA produces all SOA within 100 m of the surface. The decreases in the Northern Hemisphere are largely due to lower monoterpene emissions as shown by comparisons to OLDSOA-MEGAN2. Also notable is the increase in the surface concentration and burden of SOA around Australia due to the addition of isoprene as a precursor of SOA.

The new SOA scheme shows an improvement in simulating the mean organic mass concentrations at the IMPROVE sites, although there is a small underestimation of the annual mean organic mass. For the EMEP sites, the performance is slightly degraded, but since the process description in OLDSOA is clearly flawed, the reduced performance is more likely to result from lack of other sources. It is likely that the reason for the reduction lies in the missing production of SOA from anthropogenic sources, reactions with nitrate radical, other oxidant concentrations or even problems due to resolution.

The annual cycle at the IMPROVE stations is more realistic than before with a flat distribution comparable to the observations. At the EMEP stations the annual cycle is largely unchanged, but the underestimation of concentrations has increased. Regardless, for IMPROVE the new treatment removes a clear overestimation during the summer improving the overall SOA description.

The observations of organic mass concentrations depicted in Fig. 5 of Hodzic et al. (2016) show a different behaviour than here with a peak in the summer for IMPROVE and a flat seasonal cycle for EMEP, which is not seen in the observations here. In our model we reproduce a flat seasonal cycle for IMPROVE and EMEP, and their model reproduces the peak in the summer for IMPROVE and somewhat flat seasonal cycle for EMEP. However, their analysis included earlier periods from 2005-2008 for IMPROVE and 2002-2003 for EMEP. Nevertheless, their model with a volatility basis set (VBS) approach is more sophisticated than the scheme described here, but clearly shows that the inclusion of more complex chemistry produces better agreement with observations. Therefore, the implementation of a VBS or another more sophisticated SOA scheme into TM5 could be investigated in the future.

### 3.5 Satellite and ground-based remote sensing

In this section we compare the modelled AOD to the remote sensing observations which are described in Section 2.5.

### 3.5.1 MODIS

Figure 13a shows the difference in the annual mean collocated AOD from the NEWSOA simulation modelled AOD fields compared to MODIS (see Fig S6 for seasonal difference). The global annual mean AOD is underestimated for both the NEWSOA and OLDSOA scheme. However, in NEWSOA the global annual mean AOD is improved by 0.01 from 0.12 to 0.13, which is still 0.02 lower than the mean MODIS value (0.15), but regionally both underestimates and overestimates occur. The area weighted normalized mean bias is improved in NEWSOA to -16 % compared to -20 % in OLDSOA. The simulated AOD is underestimated in large areas over oceans in the NEWSOA and OLDSOA model runs, however the SOA scheme was not expected to impact these regions. Over land AOD is mainly increasing causing regions of increased overestimation and decreased underestimation (e.g. China and Australia). Only in some regions of Canada, Finland and Russia AOD is decreasing in an area with existing low biases (see Fig. S7 for change from OLDSOA to NEWSOA). In Russia the underestimation east of Finland has been reduced but a negative bias remains. This is expected since the wildfire emissions in this area in summer of 2010 are difficult to reproduce even with high resolution model simulations (Palacios-Peña et al., 2018). The underestimation in central Africa is reduced, but it still remains large. The strong increase in production and burden of SOA north of the Congo region changes the underestimation to a slight overestimation. In the outflow region from Africa towards South America the model has improved, but some underestimation remains. However, this is more related to dust AOD than SOA. In the Amazon region AOD is improved with NEWSOA compared to OLDSOA. Since this underestimation is mostly caused by biomass burning emissions in the Southern Hemisphere dry season (September-October-November) we expect this bias to remain with the improved SOA scheme. To summarize, the annual mean modelled AOD collocated to MODIS has increased relatively evenly. The global mean AOD is improved. However, already existing overestimations are increased in NEWSOA.

### 3.5.2 Advanced Along Track Scanning Radiometer (AATSR)

Figure 13b shows the collocated annual mean AOD difference between NEWSOA and AATSR (see Fig S8 for seasonal difference). The collocated annual mean AOD is increased by 0.01 from 0.12 to 0.13, which means that the observed global annual mean is underestimated by 0.01. Overall, the modelled AOD in Europe and Northern Russia is close to the retrieved AOD, although the wildfire region in Russia is underestimated similarly to MODIS. Over the oceans the model simulates AODs close to the retrieved values, while showing overestimations in regions where the comparison to MODIS shows agreement (e.g. subtropical Atlantic, subtropical Pacific and subtropical Indian Ocean). In contrast to the patterns of the comparison to MODIS, the AOD over Australia in NEWSOA is underestimated compared to AATSR. Furthermore, the AOD compared to AATSR over South America is underestimated as in the comparison to MODIS, although the underestimation is slightly more pronounced. However, AATSR overestimates over bright surfaces (Che et al., 2018), which means that an underestimate is to be expected in such areas, e.g. Saharan Desert, Australia.

### 3.5.3 AERONET

Figure 14 shows the comparison of AOD measured by AERONET and modelled AOD in NEWSOA (a), and OLDSOA (b). Fig. 14c shows the hemispheric seasonal cycles of AOD in AERONET observations, in NEWSOA and OLDSOA. Additionally, Fig. 14d presents a map showing the stations and their regional grouping for the statistics in Table 5. In general the simulations show similar behaviour where they overestimate the low AOD and underestimate the high AOD observations. However, in NEWSOA the mean AOD across all stations (see last row of Table 5 for values across stations) improves to 0.184 with a low bias of 0.003 from 0.176 in OLDSOA. Furthermore, the correlation coefficient (R) increases by 0.009. Visually both the NEWSOA and OLDSOA simulations show a very similar deviation from the observations.

Figure 14c shows the hemispheric seasonal cycles of the mean AOD at the AERONET stations comparing the measurements and the two simulations NEWSOA and OLDSOA. In general both simulations reproduce the observed annual cycle reasonably well. It is evident that the new SOA scheme is increasing the AOD throughout the year. However, in the northern hemisphere the observed AOD has a peak in March, but in both simulations this peak seems to be delayed by one month. The local minimum in May is not seen at all in the simulations. NEWSOA reproduces the AOD from June and July almost exactly. The small peak in August obtained with NEWSOA is closer to the observed value but is still a bit too low. The downward trend during fall (Sep-Oct-Nov), is well produced in both simulations. Even though trend is reproduced the AOD is still clearly overestimated in both simulations for September and December. In the Southern Hemisphere the highest AOD during August and September is severely underestimated in both simulations. However, NEWSOA has improved AOD during these months. At this time of the year there are strong biomass burning emissions in Brazil and southern Africa, which might be too low in the modelled emissions (Pan et al., 2020). This might explain at least part of the underestimation.

Figure 15a shows the change in AOD between OLDSOA and NEWSOA as collocated to AERONET AOD. The AOD at many stations is still underestimated with NEWSOA as shown in Fig. 15b. It shows that the AOD is increasing at most stations causing improvement at some locations and degradation at other locations. All but few stations show a larger AOD in NEWSOA than in OLDSOA, even in the boreal region where the production of SOA is reduced compared to OLDSOA. This supports the finding that the new SOA scheme is affecting particles that are more relevant for AOD. This implies that in the boreal region regardless of the lower SOA production we see an increase in AOD when the SOA is distributed more realistically onto the particles. It is noteworthy that the AOD in the Amazon has improved, but some bias remains. In Canada the decrease in burden of SOA does lead to reduction of the AOD, but the AOD is still overestimated. In central and northern Europe however, the simulated AOD has improved and is now near the observed AOD, even though the burden of SOA in the boreal region has been reduced. A comparison of biases in AERONET (Fig. 15b) and satellite retrievals (Fig. 13) — note that the color scales do not match — shows qualitative agreements between the two sets of observations at most locations and times. Over the Amazon and Southeast Asia both satellites show a decrease in AOD as do the AERONET sites. Over Europe and the US both satellite instruments and AERONET show small or no change. It is noteworthy that in Australia where satellite retrievals do not agree, the AERONET comparison shows a similar behavior as for the MODIS instrument.

The regional statistics between modelled and observed AOD at AERONET stations are presented in Table 5. The increase of AOD varies by an order of magnitude from 0.003-0.030 in the chosen regions, with the highest increase in South America (0.03). In half of the regions the new SOA scheme induced an increase in AOD that decreases the normalized mean bias (NMB) with a maximum improvement of 10.2 percentage points in South America. However, in North America, Australia, Siberia and Europe the NMB has increased, resulting in a decrease in performance compared with the OLDSOA. In Australia the increase in AOD causes a small underestimate to become an overestimate of 0.017. However, there the correlation coefficient (R) has increased to 0.499 from 0.356, and there the impact on correlation is the highest. While the NMB across the stations has improved, root mean square error (RMSE) and R at most regions show very little change. As stated earlier, it is noteworthy that the large decrease in SOA concentration at the surface in the US is not reflected in the AOD. Actually the AOD has increased in all analyzed regions, which suggests that the increase in concentrations of smaller particles are more relevant for the AOD. This is probably due to reduced growth to accumulation mode and more even distribution of the SOA mass in NEWSOA.

## 4 Discussion

The new scheme shows both performance improvements and degradation, which we will discuss in this section. In terms of SOA production the NEWSOA simulation is more in line with existing literature than OLDSOA (and the sensitivity simulation OLDSOA-MEGAN2, which gives even lower production than OLDSOA). Even with similarly low SOA production in NEWSOA and OLDSOA-MEGAN2 at high northern latitude, the global burden of SOA is increased in NEWSOA compared to either of the OLDSOA simulations. However, while there is no absolute measure of the SOA burden available, it is underestimated compared to the best estimates from Spracklen et al. (2010). This is expected because the model does not include SOA formed from anthropogenic VOC emissions or via oxidation by $NO_3$ which will be subjects for a future study. Particle concentrations around the globe are mostly increased due to an additional organic NPF scheme which provides reasonable particle formation in the boundary layer as shown in Sect. 3.2. However, in cold conditions without organic sources in Antarctica, the modelled particle concentrations are decreased and the model performance is degraded as shown in Sect. 3.3. This indicates that the effect due to the parameterised early growth of particles to 5 nm diameter should be studied in more detail. In the NEWSOA simulation the annual mean surface concentrations of organic mass in PM2.5 and PM10 are clearly improved in the US. Furthermore, there the annual cycle is corrected in NEWSOA to reproduce the observed seasonal cycle while the MEGANv2.1 emissions without the updated SOA formation in sensitivity simulation OLDSOA-MEGAN2 only affect the magnitude of concentrations. While in the US the performance is improved, in Europe the performance is degraded using the new scheme. The behaviour at the European sites would require an in-depth study to see whether the anthropogenic sources are the reason for the underestimation and the previously good agreement using OLDSOA is for the wrong reason. Globally, the modelled AOD is improved compared to satellite and in-situ observations, however, the model performance varies regionally. In the tropics AOD is increased in NEWSOA which results in a strong increase in SOA formation. However, this leads to improved performance in areas of underestimation in OLDSOA, and degrading performance in areas of overestimation. In the

northern high latitudes AOD is increasing even with a decrease in the production of SOA, which indicates that more SOA mass is distributed onto particles in the optically relevant size range by using the process-based approach in NEWSOA.

## 5 Conclusions

We have implemented a new scheme for online production of SOA from oxidation of monoterpene and isoprene together with a new particle formation mechanism depending on the ELVOC concentration. We have run the model with 1-hourly output for one year with emissions set to 2010 levels using the old and new schemes. The two main simulations have been compared to each other and a sensitivity simulation as well as to in-situ and remote sensing observations. Surface particle number concentrations were compared to observations available from the EBAS database. Surface concentrations of total organic aerosol mass were evaluated against measurements from the EMEP and IMPROVE surface networks. Aerosol optical depths were compared to retrievals from the AERONET sun-photometer network, as well as satellite observations from the AATSR and MODIS instruments.

The global production of SOA mass was increased by 35 % to 52.5 Tg(SOA) with an increase by 1.3 days in SOA lifetime (to 9.1 days), mainly due to increased concentrations of SOA in the upper troposphere. This is caused by more SOA being produced aloft in the atmosphere and a more realistic description of condensation of SOA precursors onto particles. The new lifetime is similar to the average from the AeroCom model ensemble (Tsigaridis et al., 2014). The SOA production and burden increase is rather uniform except for northern high latitudes where the production and burden are decreased compared to the old scheme. This is caused mainly by lower emissions of monoterpenes in this area in MEGANv2.1.

The introduced organically enhanced NPF scheme from Riccobono et al. (2014) induced a 100-fold increase in mean surface NPF rate. Despite this strong change in NPF rate at the surface, the mean total particle concentration is increased by only 190 cm$^{-3}$ (14 %) over the 27 EBAS stations, transforming a low NMB of -7.2 % to a high bias of 5.4 %. Although the new NPF scheme and SOA mechanisms are able to reproduce the nucleation and growth events, perhaps the concentration of CCN-sized particles from new particle formation are underestimated due to numerical issues, and too low NPF rates, and concentrations of condensable vapours for particle growth.

The simulated surface concentrations of organic mass in PM2.5 are improved especially in the US with NMB decreasing from 178 % to -18 %. A more realistic condensation of SOA mass onto particles removes the strong overestimation obtained with the old scheme, leading to a small underestimation. For EMEP sites in Europe, the new method leads to decreased performance for both PM2.5 and PM10 with NMB reduced from -7 % to -26 % (PM2.5) and -3 % to -17 % (PM10). For both EMEP and IMPROVE an underestimation is expected as only biogenic formation of SOA is considered from OH and O$_3$ oxidation of isoprene and monoterpenes, while missing anthropogenic sources and other oxidants such as NO$_3$ and differences in conversion factor from carbon to organic mass.

The global annual mean AOD collocated to MODIS and AATSR increases from OLDSOA to NEWSOA by 0.01, which brings the simulated annual mean AOD closer to observations in both cases. However, since the increase is quite uniform, regionally the change is improving areas with underestimations and degrading areas with overestimation. Since we see an

increase in AOD also at high latitudes, where the SOA production and OA burdens are decreased, the influence of particles affecting radiation is stronger. The increase could be attributed to humidity effects, but both the NEWSOA and OLDSOA simulations use the same humidity while OA does not affect water uptake. Therefore, the change in AOD results mainly from differences in the size distribution. Because the condensation of SOA in NEWSOA is physically described the particle

concentrations in the optically important size range are higher than in OLDSOA. Similarly satellite instruments, the simulated AOD collocated to AERONET measurements increases by 0.007. Even regionally the changes are small, with the strongest improvement in South America, where the bias changes from -32.7 % to -22.5 %, but also a deterioration in Australia from -4.0 % to 23.3 %. The RMSE and correlation across AERONET network improve slightly while regionally there is improvement and degradation.

Although the model shows an improvement in many aspects, especially regarding organic mass concentrations in the US, for example at EMEP sites there is a decrease in performance. As it is commonly the case, a more detailed process description can initially lead to reduced performance due to error compensation in the simpler description. Further development is needed targeting the identified model deficiencies. Some future lines of study could focus on 1) missing anthropogenic SOA sources, 2) additional precursor compounds (e.g. sesquiterpenes and anthropogenic gases) and oxidants (e.g. $NO_3$), 3) transport and

removal of gas-phase SOA precursors (e.g. ELVOCs and SVOCs), 4) interactive emission of precursor gases from a the dynamic vegetation model.

*Code and data availability.* The TM5-MP code for Version 1.2 used in this work is available at Zenodo (10.5281/zenodo.5559644). However, we urge interested researchers to get access to the gitlab repository to download the latest version. Access can be obtained by emailing Philippe Le Sager (sager@knmi.nl). Post-processing and plotting scripts used for creating figures 2, 4-8, and 10-15 are available on

github (https://github.com/tommibergman/gmdd-tm5-soa and on zenodo: doi://10.5281/zenodo.5561959). Simulation datasets are available at https://doi.org/10.23729/b1eb8b20-83c4-4dc6-a1fd-05dc13fe42c3.

Observational data for CPC and EMEP data from EBAS are available for download from the EBAS database at http://ebas.nilu.no/ (last access: EMEP 27.7.2017, CPC 20.1.2020) , Norwegian Institute for Air Research, 2015).

IMPROVE data are available for download from http://views.cira.colostate.edu/fed/QueryWizard/Default.aspx (last access: 11.1.2018),

AERONET data can be downloaded using the Aerosol Robotic Network download tool https://aeronet.gsfc.nasa.gov/cgi-bin/webtool_aod_v3 (last access: 22.2.2016).

MODIS data are available for download from the Atmosphere Archive and Distribution System (LAADS) https://ladsweb.modaps.eosdis.nasa.gov/search/ (last access: 3.3.2017).

AATSR data are available for download from Centre for Environmental Analysis (CEDA) http://data.ceda.ac.uk/neodc/esacci/aerosol/data/AATSR_SU/L2

(last access: 17.1.2017) .

## Appendix A:  Parametrisation of growth from nucleation to 5 nm

As shown by Kerminen and Kulmala (2002) in Eq. 21, the growth rate (in $\mathrm{nmh^{-1}}$) due to gas-phase compound $i$ can be approximated

$$\mathrm{GR}_i \approx \frac{3.0 \times 10^{-9}}{\rho_{\mathrm{nuc}}} \sum_i \bar{c}_i M_i C_i \qquad \text{(A1)}$$

Here the nuclei density $\rho_{\mathrm{nuc}}$ is assumed to be 1 $\mathrm{gcm^{-3}}$. The molecular speed $\bar{c}_i$ is calculated online depending on the environment and gas-phase properties of either SA or ELVOC. Molecular weight $M_i$ for ELVOC is 248 $\mathrm{gmol^{-1}}$ and for SA 98 $\mathrm{gmol^{-1}}$. The gas-phase concentration $C_i$ for both gases is calculated online.

By combining Eq. 11 and 13 from (Kerminen and Kulmala, 2002) we can calculate the fraction $F$ of particles surviving to diameter $d_p$ using the growth rates ($\mathrm{GR}_{SA}$,$\mathrm{GR}_{ELVOC}$), reduced condensation sink CS' of existing aerosol population and semi-empirical proportionality factor $\gamma$. Here we calculate CS' following Kulmala et al. (2001) and $\gamma$ following Kerminen and Kulmala (2002).

$$F = \exp\left[ \left( \frac{1}{\rho_p} - \frac{1}{\rho_{\mathrm{nuc}}} \right) \frac{\gamma \mathrm{CS'}}{\mathrm{GR}_{SA} + \mathrm{GR}_{ELVOC}} \right] \qquad \text{(A2)}$$

This fraction is calculated and used in two phases during the NPF parameterisation. First, to calculate the fraction surviving to 1.7 nm from binary nucleation. Secondly, to calcluate the fraction surviving to 5 nm from combined $J_{1.7}$ from binary nucleation and Riccobono new particle formation.

Volume fractions ($V_f r$) for SA and ELVOC ($j$) in the produced 5 nm particles are assumed to be proportional to the cube of their growth rates.

$$V_{fr,j} = \frac{\mathrm{GR}_j^3}{(\mathrm{GR}_{\mathrm{SA}} + \mathrm{GR}_{\mathrm{ELVOC}})^3} \qquad \text{(A3)}$$

Which are then normed to account for all mass in the particles:

$$V_{fr,SA} = \frac{V_{fr,\mathrm{SA}}}{V_{fr,\mathrm{SA}} + V_{fr,\mathrm{ELVOC}}} \qquad \text{(A4)}$$

Rest of the volume is then assigned to the remaining compound

$$V_{fr,\mathrm{ELVOC}} = 1 - V_{fr,\mathrm{SA}}. \qquad \text{(A5)}$$

In case of insufficient amount of compound $i$ we recalculate the volume fractions so that the other gas phase compound is assumed to produce the remaining growth to 5nm.

$$V_{fr,i} = \frac{C_i M_i}{N_{5\,\mathrm{nm}} v_{tot} R_a \rho_i \Delta t} \qquad \text{(A6)}$$

Here $C_i$, $M_i$ and $\rho_i$ are the concentration, molecular weight and density of $i$, respectively. $R_a$ is the Avogadro number, $v_{tot}$ is the volume of the 5 nm particle. $\Delta t$ is the length of the timestep. The volume fraction for the other compound is then calculated using A5

Furthermore, if both gas-phase compounds are too low to uphold the formation rate at 5 nm recalculate the formation rate based on the available gas-phase compound concentrations.

$$J = \frac{1.0}{R_a v_{\text{tot}} \Delta t} \left( \frac{C_{\text{SA}} M_{\text{SA}}}{\rho_{\text{SA}}} + \frac{C_{\text{ELVOC}} M_{\text{ELVOC}}}{\rho_{\text{ELVOC}}} \right) \tag{A7}$$

The $n_i$ ($i$=SA,ELVOC) is removed from the available gas-phase concentration before calcalation of the condensation on the particle population.

$$n_i = \frac{V_{fr,SA} v_{\text{tot}} R_a \rho_{\text{SA}}}{M_{\text{SA}}} \tag{A8}$$

*Author contributions.* TB and RS ran the simulations. TB, RM, TvN, PLS and RS co-developed the TM5-MP model. TB, RS, RM, and TvN contributed to the experimental design and data analysis. All authors contributed in the writing of paper.

*Competing interests.* Authors declare no competing interests.

*Acknowledgements.* This work was suproted by the European Union's Horizon 2020 research and innovation programme under grant agreement no. 641816, Coordinated Research in Earth Systems and Climate: Experiments, kNowledge, Dissemination and Outreach (CRESCENDO). RS acknowledges financial support from the Strategic Research Area MERGE (Modeling the Regional and Global Earth System - www.merge.lu.se). Authors acknowledge the ACTRIS network for the aerosol size distribution data used for new particle formation analysis at Birkenes, Harwell, Hyytiälä, Mace Head Vavihill and Waldhof. Authors thank ECMWF (European Centre for Medium range Weather Forecasts and CSC - IT Center for Science for computational resources.

Analysis was done mainly by following open source tools: Python (Perez et al., 2011; Oliphant, 2007; Millman and Aivazis, 2011), Xarray (Hoyer and Hamman, 2017), MatPlotlib with basemap (Hunter, 2007), SciPy (Jones et al., 2001–), NumPy (Oliphant, 2006–), CIStools (Watson-Parris et al., 2016)

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

Table 3. Global annual SOA budget and precursor emissions in the two simulations NEWSOA and OLDSOA. Literature values are shown in columns marked with letters: J for Jokinen et al. (2015), S for Spracklen et al. (2011), T for Tsigaridis et al. (2014), K for Kanakidou et al. (2005), and G (Goldstein and Galbally, 2007) St for (Stadtler et al., 2018).

| | NEWSOA | OLDSOA | OLDSOA-MEGAN2 | J | S | T | K | G | St[c] |
|---|---|---|---|---|---|---|---|---|---|
| Total emission of monoterpenes [Tg yr⁻¹] | 95.5 | 127[a] | 95.5 | 86 | | | 127.4 | | |
| of which biomass burning [Tg yr⁻¹] | 1.07 | – | | | | | | | |
| Total emission of isoprene [Tg yr⁻¹] | 572.1 | – | | | | | | | |
| of which biomass burning [Tg yr⁻¹] | 0.41 | – | . | | | | | | |
| Burden [Tg] | 1.31 | 0.85 | 0.70 | – | 1.81 | 1.0 (0.3–2.3) | | | 1.4[c] |
| Total SOA production [Tg yr⁻¹] | 52.5 | 39.9 | 30.9 | 27 | 140[b] (50–380) | 36.7 (12.7–120.8) | 12–70 | 140-910 (in C) | 138.5[c] |
| contribution of ELVOC [Tg yr⁻¹] | 7.3 | – | | | | | | | |
| from isoprene [Tg yr⁻¹] | 0.71 | – | | | | | | | |
| from monoterpenes [Tg yr⁻¹] | 6.56 | – | | | | | | | |
| contribution of SVOC [Tg yr⁻¹] | 45.2 | – | | | | | | | |
| from isoprene [Tg yr⁻¹] | 23.9 | – | | | | | | | |
| from monoterpenes [Tg yr⁻¹] | 21.3 | – | | | | | | | |
| Wet Deposition [Tg yr⁻¹] | 52.0 | 39.2 | 30.4 | | 47.9 (12.4–113.1) | 133.6[c] | | | |
| Dry Deposition [Tg yr⁻¹] | 0.56 | 0.74 | 0.53 | | 5.7 (1.4–14.5) | 4.4[c] | | | |
| Lifetime [days] | 9.1 | 7.8 | 8.2 | | 8.2(2.4–14.8) | 3.7[c] | | | |

[a] Used only for calculating the SOA production.

[b] Including SOA sources other than monoterpene and isoprene.

[c] Including only isoprene source for SOA.

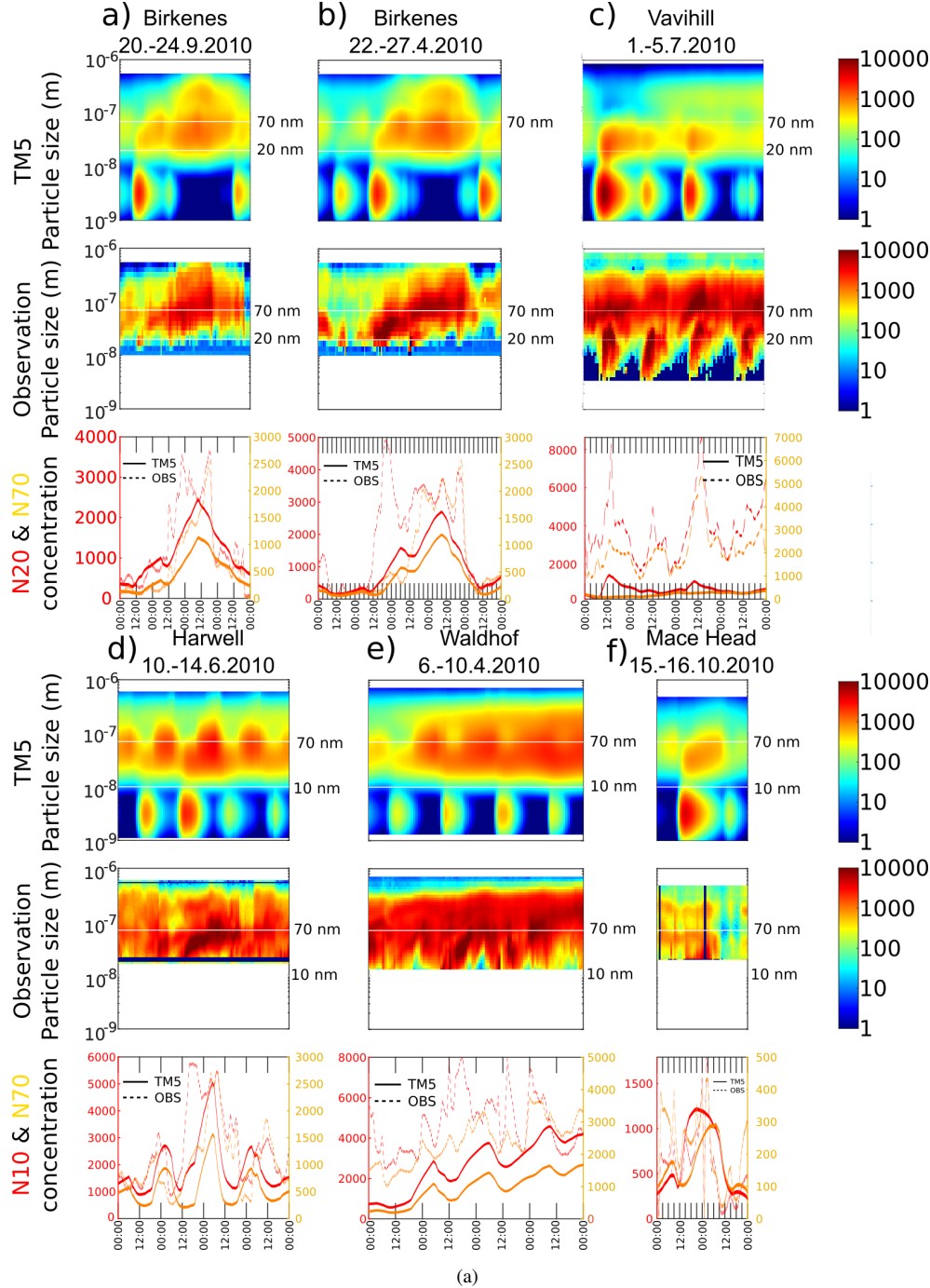

**Figure 9.** Comparison of TM5 aerosol size distribution (top row) against selected events in 5 observation sites (middle row) during 2010: Birkenes (a,b), Vavihill (c), Harwell (d), Waldhof (e) and Mace Head (f). The bottom row shows integrals over the size distribution for N10 ($d_p > 10$), N20 ($d_p > 20$, in red lines) and N70 ($d_p > 70$ nm, in yellow). N10, N20 and N70 are shown in the size distribution plots as white lines.

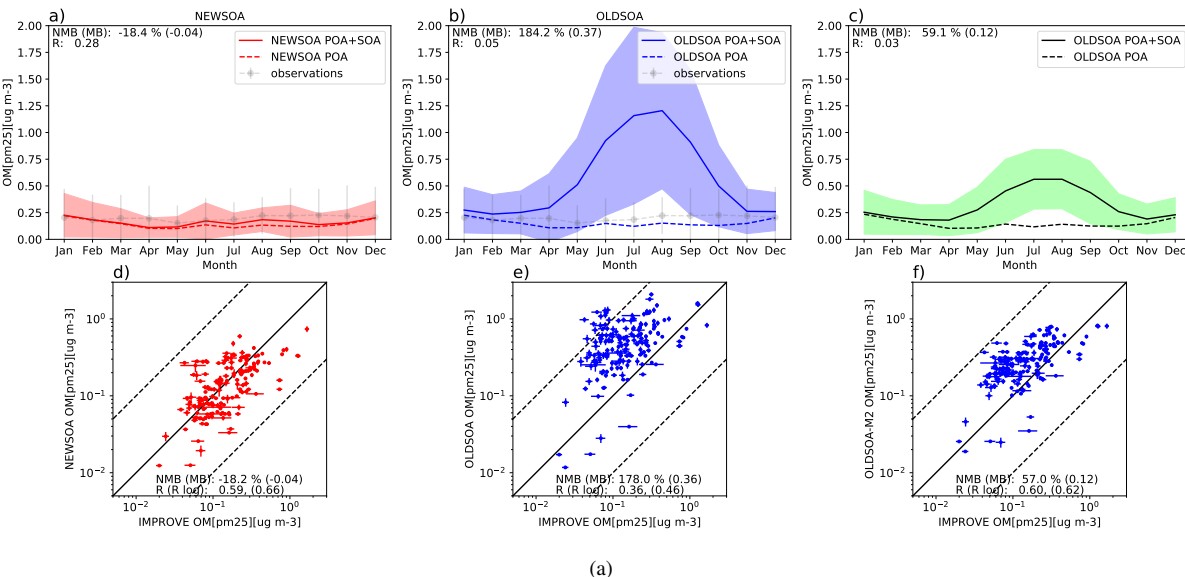

(a)

**Figure 10.** Organic mass concentrations at IMPROVE sites for the three simulations. Top row shows the annual cycle in monthly mean organic mass concentrations for 2010 at the surface NEWSOA (a) , OLDSOA (b) and OLDSOA-MEGAN2 (c). The modelled concentrations are collocated to observational data temporally and spatially. The shading indicates the standard deviation from the mean. Grey dots show the observed mean with standard deviation as vertical lines. The bottom row has scatterplots of annual mean concentration of organic mass in PM2.5 particles at IMPROVE sites in the NEWSOA (d), OLDSOA (e) and OLDSOA-MEGAN2 (f) simulations. In the scatterplots the height and width of line of the sign shows the standard error of the mean for modelled and observed concentrations, respectively.

**Table 4.** Modelled and observed annual mean OA concentrations, normalized mean bias (NMB) and correlation (R) for monthly mean organic mass concentration for EMEP and IMPROVE stations.

| Stations | Size | OBS [$\mu g\,m^{-3}$] | MODEL [$\mu g\,m^{-3}$] | | NMB | | R | |
|---|---|---|---|---|---|---|---|---|
| | | | NEWSOA | OLDSOA | NEWSOA | OLDSOA | NEWSOA | OLDSOA |
| EMEP | PM25 | 3.38 | 2.56 | 3.23 | -26 % | -7 % | 0.64 | 0.80 |
| EMEP | PM10 | 2.98 | 2.24 | 2.57 | -17 % | -3 % | 0.81 | 0.81 |
| IMPROVE | PM2.5 | 0.20 | 0.15 | 0.56 | -18 % | 184 % | 0.28 | 0.05 |

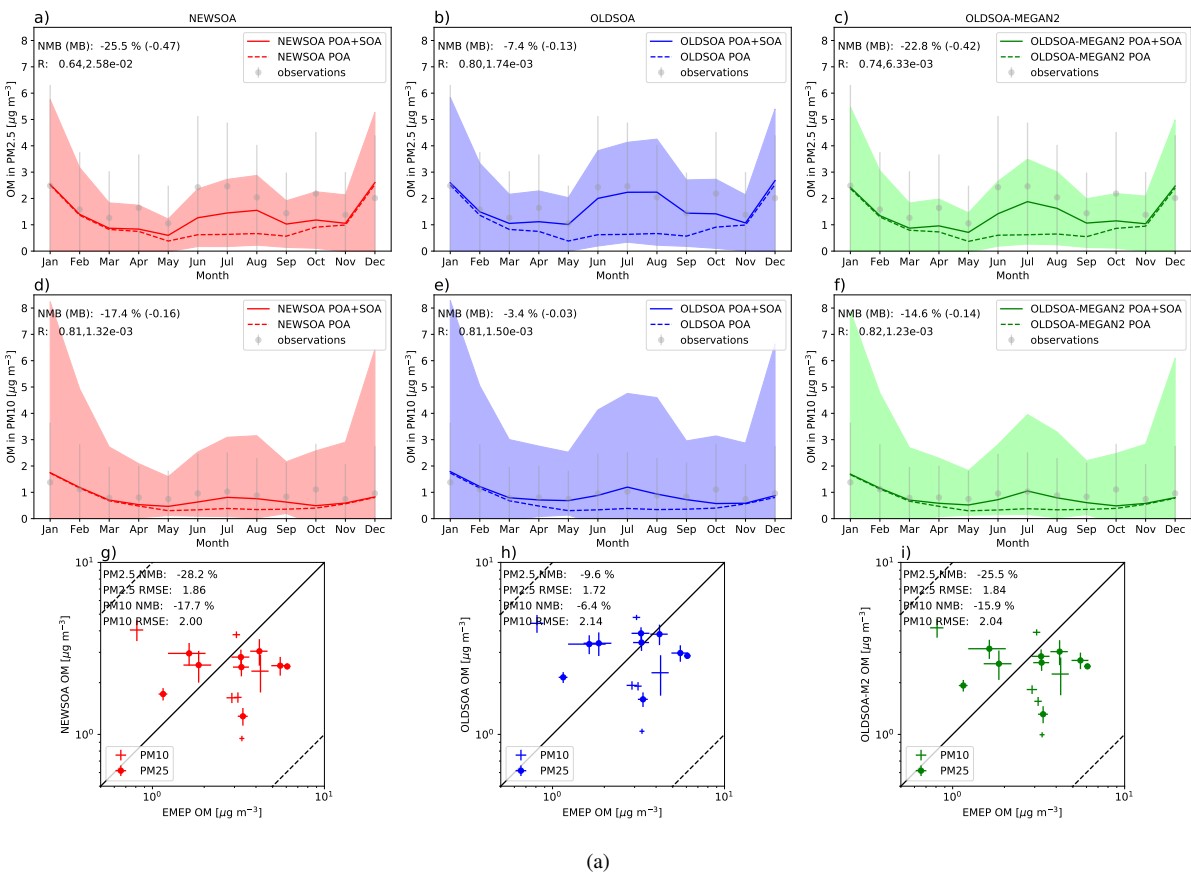

(a)

**Figure 11.** Organic mass concentrations at EMEP sites for the three simulations for PM2.5 and PM10. Top row shows the annual cycle in monthly mean organic mass PM2.5 concentrations for 2010 at the surface NEWSOA (a), OLDSOA (b), and OLDSOA-MEGAN2 (c). The middle row shows the annual cycle in monthly mean organic mass PM10 concentrations for 2010 at the surface NEWSOA(d), OLDSOA (e) and OLDSOA-MEGAN2 (f). The modelled concentrations are collocated to observational data temporally and spatially. The shading indicates the standard deviation from the mean. Grey dots show the observed mean with standard deviation as vertical lines. The bottom row has scatterplots of annual mean concentration of organic mass in PM2.5 (◇ with +) and PM10 (+) particles at EMEP sites in the NEWSOA (g), OLDSOA (h) and OLDSOA-MEGAN2 (i) simulations. In the scatter plots the height and width of line of the sign shows the standard error of the mean for modelled and observed concentrations, respectively.

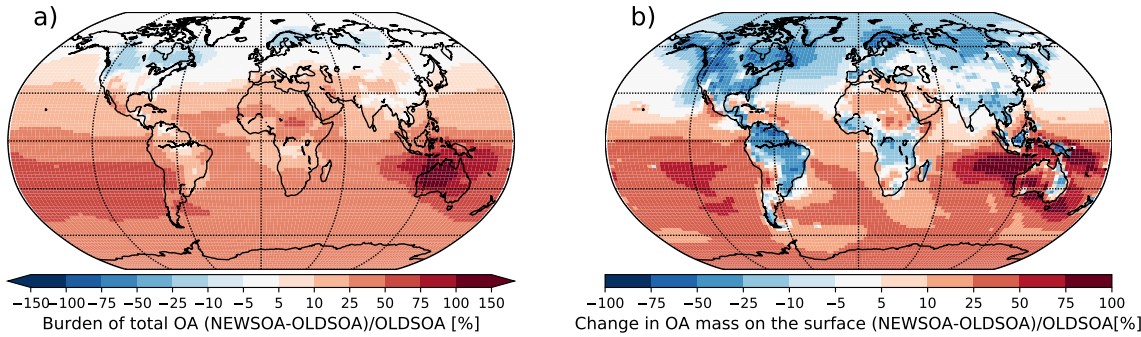

**Figure 12.** Annual mean difference between NEWSOA and OLDSOA in a) OA mass burden in the atmosphere and b) OA mass concentration at the surface.

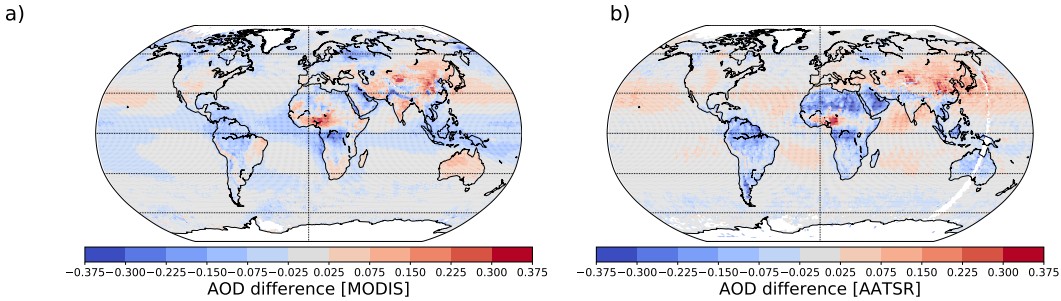

**Figure 13.** Annual mean AOD difference between the NEWSOA and a) MODIS and b) AATSR retrievals. Blue and red color indicate underestimation and overestimation, respectively. White areas indicate missing data in both of the retrievals.

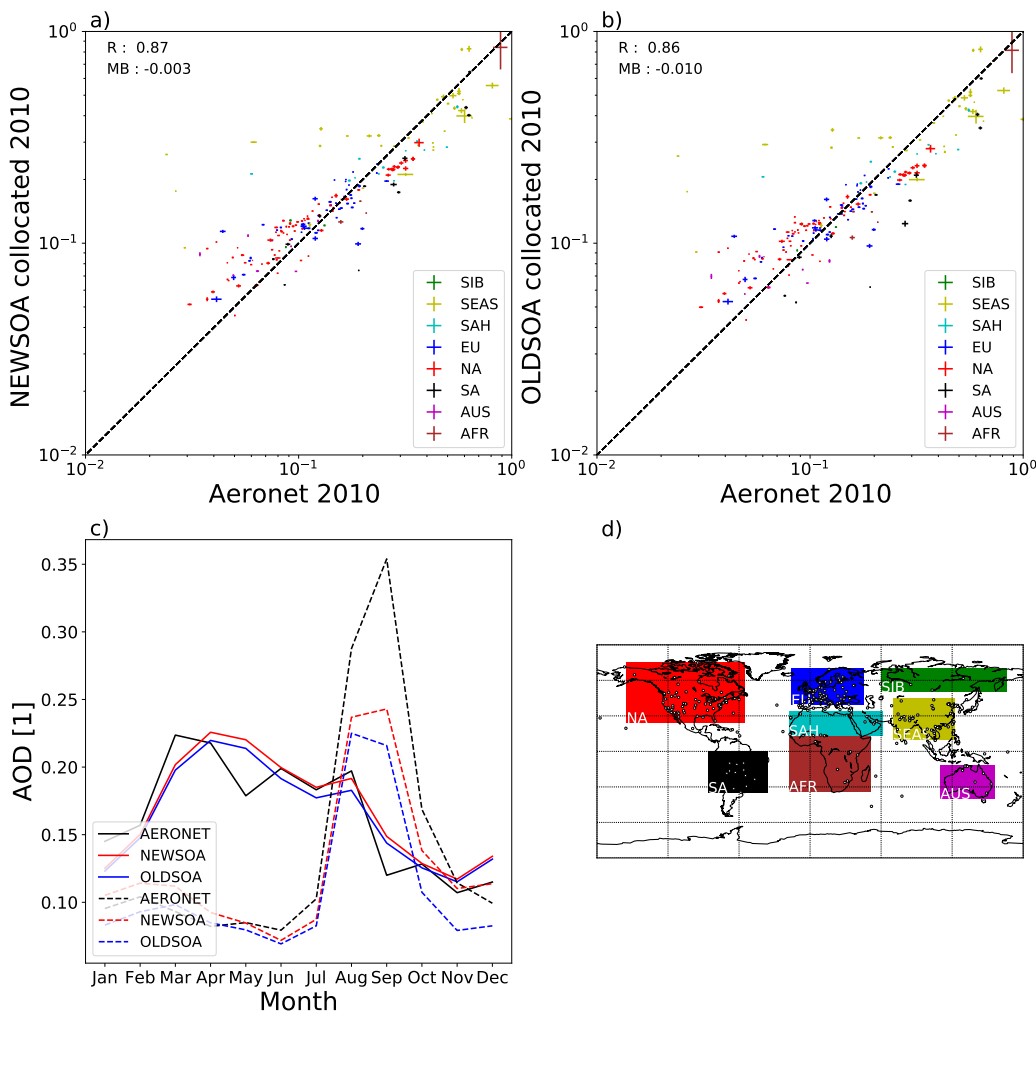

(a)

**Figure 14.** AOD at 550 nm for the year 2010 from AERONET compared with collocated values from the simulations. The top panels show scatter plots of the retrieved and simulated annual mean values for (a) NEWSOA and (b) OLDSOA. Colors indicate different geographical locations. Plotted values show standard error of the mean for both observed and modelled values, but errors are very small. Panel c) shows the hemispheric seasonal cycle of the mean AOD across all AERONET stations with derived AOD in black, NEWSOA in red and OLDSOA in blue (northern hemisphere: solid line; southern hemisphere: dashed line) . Panel d) shows the locations of the observation stations and the areas used for grouping. Absolute mean bias MB and correlation coefficient R (logarithmic R in parentheses) across all stations are annotated in panels a) and b).

**Table 5.** Annual mean AERONET AOD at 550nm, the normalized mean bias (NMB), correlation coefficient (R) and root mean square error (RMSE) for simulations NEWSOA and OLDSOA . The number of stations is denoted with N. The regions listed in the first column are indicated in Fig. 14.

| Region | N | AOD | | | NMB in % | | R | | RMSE | |
|---|---|---|---|---|---|---|---|---|---|---|
| | | AERONET | NEWSOA | OLDSOA | NEWSOA | OLDSOA | NEWSOA | OLDSOA | NEWSOA | OLDSOA |
| SIB | 3 | 0.104 | 0.115 | 0.112 | 11.1 | 7.7 | 0.401 | 0.472 | 0.023 | 0.020 |
| SEAS | 37 | 0.392 | 0.372 | 0.364 | -5.1 | -7.1 | 0.664 | 0.656 | 0.170 | 0.172 |
| SAH | 12 | 0.291 | 0.257 | 0.248 | -11.9 | -15.0 | 0.817 | 0.818 | 0.097 | 0.101 |
| EU | 49 | 0.134 | 0.140 | 0.135 | 4.1 | 0.7 | 0.808 | 0.819 | 0.031 | 0.029 |
| NA | 80 | 0.126 | 0.130 | 0.126 | 3.5 | -0.2 | 0.965 | 0.960 | 0.036 | 0.040 |
| SA | 12 | 0.295 | 0.228 | 0.198 | -22.5 | -32.7 | 0.934 | 0.924 | 0.101 | 0.127 |
| AUS | 7 | 0.074 | 0.091 | 0.071 | 23.3 | -4.0 | 0.499 | 0.356 | 0.031 | 0.027 |
| AFR | 5 | 0.321 | 0.285 | 0.266 | -11.3 | -17.3 | 0.997 | 0.997 | 0.043 | 0.060 |
| All | 258 | 0.187 | 0.184 | 0.176 | 0.0 | 0.1 | 0.874 | 0.865 | 0.079 | 0.083 |

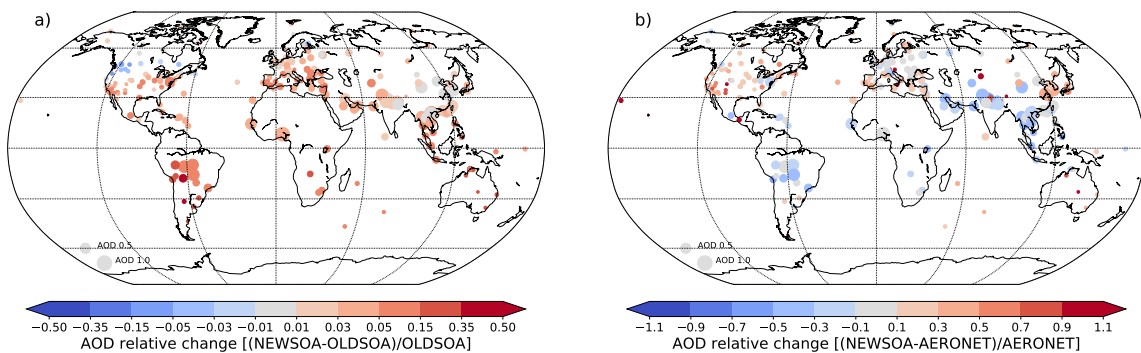

**Figure 15.** Annual relative change of AOD at AERONET stations between NEWSOA and OLDSOA (a). Red indicates increase and blue decrease compared to OLDSOA. Panel (b) shows the annual mean relative bias in NEWSOA at AERONET stations. In both panels larger markers indicate higher observed AOD in both (a) and (b).