# Peer review of "Description and Evaluation of a Secondary Organic Aerosol and New Particle Formation Scheme within TM5-MP v1.12"

_Geoscientific Model Development, 2021_

## Referee Comment (RC1)

This paper describes a new mechanism for formation of SOA and compares thee results from the new mechanism to the old model. In addition both the old and new models are compared with observations.

The new mechanism is based on measurements of molar yields of SOA from Jokinen et al 2015 and Kroll et al (2005). In addition, the ELVOC and SVOC forms of SOA are also based on Jokinen et al 2015.

There are a number of figures and tables in the supplement which are never referred to in the paper. In addition, Fig 13 in the paper does not show the figure being described in the text.

In the following I detail changes/explanations that are needed:

Please explain how you calculate OH and O3. Are these read-in?

Line 34: odd addition of numbers in "vegetation"

Line 46:  please add references to models producing up to 121 Tg/yr SOA

Line 47: While VBS schemes are "more intricate" they are essentially parameterizations. You need to state this.

Line 52:  "subsequently" is a strange word here. "In this theory"?

Line 60: remove first "of"

Line 69:  add Wang and Penner, 2009:

Wang, M. and J. E. Penner, 2009: Aerosol indirect forcing in a global model with particle nucleation, Atmos. Chem. Phys., 9, 239-260, www.atmos-chem-phys.net/9/239/2009/.

After all, this paper was published almost a full year before Merikanto et al., 2009

Lines 101-103: You do not follow the mixing of soluble and insoluble particles? This seems highly unrealistic, since, for example SO4 would almost certainly condense onto insoluble particles. Are simulating sulfate condensation only on the soluble modes? If on both types of particles, how do you separate into the modes after condensation, since sulfate and dust forming on an Accumulation particle, could, after separation, form 2 Aitken mode particles or 1 Aitken mode and 1 accumulation mode, depending on the size of the original particle after condensation.

Line 110: delete "more" in "A more detailed description", since no description is given.

Figure 1: Is there some reason that the reaction of isoprene and monoterpenes with NO3 is omitted?

From the figure the dotted line for the insoluble mode is only green for the Aitken model. So, it seems ELVOC and SVOC only condense on the insoluble Aitken mode. Why is this?

Line 139-140: does the diurnal emission variation for monoterpene at least depend on the local temperature variation? Sureely the model has this variable available. Please explain why it should not take this into account, if not and give the max/min dependence of the emission.

141: This reads like the monthly mean isoprene emission is 572.3 Tg/yr, but I know that is the annual emissions rate. I think you mean you employ a monthly varying emission with an annual mean of….

175-176: Why wouldn't SVOC partition onto Accumulation and coarse modes if OA is present? (for example, by ELVOC condensation on these modes)?

215: I gather "all other biogenic emissions" really just refers to isoprene? Why not say this?

218-225: I think a paper should give sufficient information so that the reader does not need to read other papers in order to understand. Why not add a table with all annual emissions and references? After all, the emissions are one basic element affecting the evaluation

230: I don't think you mean SOA "on" the surface. Perhaps "in the boundary layer"? or in the first level of the model.

246: The factor of 1.6 should be referenced, unless there is no reference. If basically unknown, then say so.

247: do not use "on the surface"

250: change "overpresented" to "overrepresented"

313-314: here you first mention that there is no SOA formation from isoprene in OLDSOA. This should be stated earlier when talking about the sources from MEGAN (new and old).

326-327: can you speculate why the wet deposition fraction is higher in your model? Perhaps the vertical distribution places SOA higher in the atmosphere than in the other models?

Figure 7:  Are (c) and (d) concentrations in the surface layer?

341-342: change to "using the additional particle formation"  - I had to look back to see if you also include the Vehkamki scheme for sulfate in NEWSOA

352-357:  please explain why southern ocean nucleation mode particle numbers are higher in OLDSOA than NEWSOA. After all, both these schemes include binary homogeneous nucleation

361: missing e in "increase"

365: "of of"

369-370: I do not understand why particle formation in NEWSOA isn't always larger than in OLDSOA, since the former includes RiCCO AND binary sulfate formation

371-373:  what is meant by "compensated"? this explanation needs some expansion – it is hard to follow.

384:  I think you do not mean observed N2O, since this is a gas. Also, it appears CCN particles do not reach 5000/cm3: the scale only goes to 3000/cm3, but Vavill goes to 5000/cm3. BTW if you are referring to the top 2 lines of Fig 9b, there is no scale, so one cannot tell if the observation goes to 5000/cm3.

Figure 11: in (e) and (f), I assume the 2 NMB's and RMSE's are for PM2.5 and PM10, but they are not labeled or in the caption. Also, what are the 2 numbers after R: in the upper 2 panels?

Figure 13:  the label on the graph states that both a and b are the difference between NEWSOA and OLDSOA, whereas the caption states the correct difference.

472:  Where is OLDSOA compared with MODIS and AATSR? The figure caption of figure 13 states that this figure is only NEWSOA. Also, it appears that both panels in fig 13 have most of the tropics and subtropics overestimated, so how do you estimate an underestimated AOD? Much of the paragraph after this sentence in line 472 cannot be understood based on fig. 13.

So, when you finally reference the supplemental figures, I see that the supplement Fig S10 and S11 compares OLDSOA and NEWSOA with MODIS  and AATSR. I find it strange that this is perhaps the first supplemental figure (or table) that is meant to be referenced. Why have figures and tables in a supplement if they are not referenced?

BTW, I would at least include an annual average difference of OLDSOA and NEWSOA with MODIS  and AATSR in the main text. As it stands, the main result of the data shown in the current Fig 13 was already discussed.

552: add a comma after "events"

553: explain what the "numerical issues" are. This is the first mention of this fact. Also, I would add "perhaps" since you do not actually analyze the reason for low CCN sized particles (by comparison with observations, for example)

567:  The change in AOD is ultimately from the change in particle number, but must actually be due to the change in size distribution, I think. This sentence should be expanded to remind the reader of the processes leading to the change in AOD

---

## Referee Comment (RC2)

Review of "Description and Evaluation of a Secondary Organic Aerosol and New Particle Formation Scheme within TM5-MP v1.1"

In this study the authors present and evaluate their implementation for new particle and secondary organic aerosol formation from semi and extremely low volatility monoterpene and isoprene oxidation products. The new implementations expand the functionality of the M7 aerosol module within a global chemistry transport model (TM5-MP). The authors explain the new functionalities in detail, including the updated biogenic emission inventory, the chemical conversion to SVOC and ELVOC, the new particle formation and SOA formation approaches. After introducing the changes, the authors compare the extended M7 scheme with an older version, by comparison of the simulations themselves, but also including observations. Observational datasets to evaluate the global model are surface organic mass concentrations (IMPROVE and EMEP networks), number concentrations on the surface (EBAS), satellite and ground based AOD products (MODIS, AATSR, AERONET).

The chemical conversion from isoprene and monoterpenes to SVOC and ELVOC in form of lumped species is not novel, but the extension of M7 functionalities can be applied for a number of global chemistry models and thus makes a fair contribution to the aerosol model development in general. The manuscript is generally well structured and I recommend that the manuscript should be accepted for publication after major revisions where the authors carefully addressed my comments.

**General comments:**

The TM5-MP model description lacks the spatio-temporal model resolution used for this study. In the evaluation and conclusions, the time resolution becomes clear. In section 2.4 the simulation period, and later in the text also the horizontal resolution, are mentioned. For the reader it would be easier to get the full information on spatial and temporal resolution, such as the time period for the simulations either right at the beginning in Section 2.1 or in Section 2.4 in a more coherent way.

In Section 2.3 the earlier version of TM5 is described and it is stated that it uses biogenic monoterpene emissions from MEGAN v1 and does not include isoprene emissions for SOA formation. In contrast, the new version calculates SOA from isoprene and monoterpenes, based on MEGAN v2 inventories. In section 2.4 the authors describe the simulations to be compared, one simulation using the older version "OLDSOA" and a second simulation for the same time period using the new formation schemes "NEWSOA". The difference between these two simulation configurations are not only the novel SOA and new particle formation schemes, but also the changed biogenic emissions plus, in the old scheme only monoterpene-derived SOA was included. For disentangling the contributions from the updated biogenic emissions from the contributions by the extended SOA treatment, I would suggest to add sensitivity simulations. I wonder how does the NEWSOA scheme perform compared to OLDSOA while using the same MEGAN v1 emissions? Also, how do NEWSOA and OLDSOA compare when both using MEGAN v2 emissions, even if OLDSOA does not form isoprene-derived SOA? I would not expect the difference in emissions shown in Figure 2 apply in a linear way to SOA formation. In Section 3.1 the last paragraph points into the direction. Additional sensitivity simulations with both emission inventory versions would help to understand what

difference is caused by the changed emissions and what difference is related to the novel SOA formation pathways.

In section 2.3.2 the production of ELVOC and SVOC is described. In the introduction, the authors do mention low volatility compounds (LVOC), but do not explicitly state here why they assume two products of the isoprene and monoterpene reactions (ELVOC and SVOC) and leave out the possible LVOC products. It would be helpful for the reader to know the reason for this decision.

Table 2 nicely shows the SOA budget together with some literature values. In this study, the authors differentiate between monoterpene-derived SOA and isoprene-derived SOA, a distinction which cannot be found in the literature cited in Table 2. A study focusing on isoprene derived SOA could add some more information on how to interpret the values for isoprene-derived SOA. Please consider comparing to the global isoprene-derived SOA values in Table 4 given in Stadtler et al. 2018 (https://doi.org/10.5194/gmd-11-3235-2018). I did not find a comparable study for monoterpene-derived SOA on a global scale, but it might also exist.

In Section 3.3 the authors start to compare the OLDSOA and NEWSOA simulations to observed particle number concentrations (Figure 8). Although this kind of scatter plot is a common evaluation method, the comparison shown here does not include any error or uncertainty indication, either for the observations, nor for the statistical error caused by comparing grid cells to point measurements. Swall et al. 2009 (https://doi.org/10.1016/j.atmosenv.2008.10.057) describe and discuss the incommensurability problem arising from comparison of point measurements to grid cell values. According to Swall et al. even if a perfectly performing model and measurements without observational error, the scatterplot will not show modeled and observed values on a one-to-one line. The same holds true for Figure 14 and comparison to AERONET stations, Figure 10 in comparison to IMPROVE stations and Figure 11 in comparison to EMEP stations. The authors should consider including information on uncertainty derived from the comparison approach and, if available, the observational errors. A model error would be nice-to-have, but I dare not to ask for that.

In whole Section 3 the description of the results and comparison is blended with the discussion of the model errors and limitations. The authors should consider editing the subsections in Section 3 such that the description of the results is separated from the discussion and interpretation of the model strengths and weaknesses. A clear strength of this study is the evaluation with a variety of observations, but it makes the manuscript difficult to read if the interpretation and discussion is mixed into the comparisons to the old model and observations. The authors should consider to collect all the discussion points of NEWSOA schemes' limitations and advantages and to write them into a single "Discussion" Section in a concise way.

Out of curiosity, why was it not possible to include isoprene and monoterpene oxidation by NO3 in this study? As you mention, it limits the current model performance, but should be considered in a future study. I wonder what reason was prohibiting to implement this additional oxidation reaction in the NEWSOA scheme.

Technical corrections:

In the abstract line 11 – 14 it is written "Compared to the old scheme, the new scheme
is increasing the number concentrations across the observation stations while still underestimating the
observations. The total aerosol mass concentrations in the US show a much better seasonal cycle and
removal of a clear overestimation of concentrations.", which is confusing the reader. First the authors
speak of "still underestimating" but the next sentence reads "removal of overestimation". Please
make it clearer here.

In Line 34 there is some number in the word "vegetation" "vegetat589527ion".

Line 431 and 432: Two sentences in a row starting with "However".

In Figure 13 the plot labels indicate [NEWSOA-OLDSOA], but according to the caption of Figure 13 the
annual mean difference of NEWSOA and MODIS (a), NEWSOA and AATSR (b) are shown. Please
change the label of the plots.

Line 504 it says "However, compared to OLDSOA the absolute bias across all stations increases in
NEWSOA by 0.007 to a low bias of 0.003", should the verb be "decreases"?

---

## Referee Comment (RC3)

Review of "Description and Evaluation of a Secondary Organic Aerosol and New Particle Formation Scheme within TM5-MP v1.1"

This article describe the inclusion of a more explicit parameterisation of the formation of secondary aerosol and nucleation induced by extremely low volatile organic compounds and sulphuric acid and new emissions aerosol in M7 aerosol module embedded within the global chemistry transport model TM5-MP.

The work includes a description of these processes as well as a comparison with the previous parameterisations including emission changes

The authors have included a quite extensive comparison with measurements of both aerosol mass, number concentration, and aerosol optical depth. This output is given with a high temporal resolution of output in order to catch individual nucleation events. The relatively coarse resolution of the CTM make it however make a detailed comparison with station sites.

While none of the parameterizations are new, the more physical approach  is a useful addition to aerosol modelling tools, in particular for M7 and derivatives thereof.  The article is in general well structured although sometimes it is a bit unclear when the authors describe the results from the old or the new version. A somewhat more problematic issue is that the authors often conclude that this is caused by emission changes and this is due to secondary aerosol formation. While it may be obvious for most cases I think the authors should do a sensitivity test that only show the impact of the emission changes. The assumptions on yields is another parameter that should be discussed although the emission test will probably work as a first approximation on understanding yields.

It may beyond the scope of the article and or due to missing capabilities of the model I am wondering if the authors also considered comparing the Ångstrøm component in particular in light of the focus of aerosol size changes from the new parameterisaton.

The paper is suitable for GMD and I recommend that the papers should be published after the manuscript provides a better description on the effect of emission changes versus the microphysical / chemcial changes in the model. I have also added some minor issues below

Minor / technical issues.

The introduction splits the organic compounds between ELVOC, LVOC and SVOC, Later on it only use ELVOC and SVOC. My usual interpreation is that SVOC s reserved for thos compounds that can commonly re-evaporate. As I understand none of the VOCs described in this work is re-evaporating. I think this should be stated explicitly if the authors continue to use SVOC.

Abstract line 11. "clearly lower as is"  new versus old ?

Line 32: Missing in first volatility word

Line 34. Numbers inside "vegetation"

Line 39: Two "rathers" are rather too much. Can one of them e.g. the first be replaced by a number / fractional estimate.

 Line 76: "between" → "divided between" ?

Line 93, but this has also consequences for the remaining manuscript. Does "trace gases "include OH? If not the description of chemistry must include information on the OH concentration used in the model.
If OH is part of the chemistry. Will the SOA chemistry have noticeable impact on the OH concentration?
Figure 1: No evaporation of SVOC?

Line 121. Are isoprene and monoterpene emitted at ground level or follow the height distribution of the oldSOA.

Line 143: Unclear. "geographical distribution ..." As far as I understand the model reads regridded monthly mean emissions from a specific MEGAN set-up. TM5 does not do anything to the emissions themselves e.g. related to meteorological fields? so it is just how the emission look like from the model (except for the diurnal cycle)

Line 156. Choice of yield looks a bit arbitrary. Tuning ? A sensitivity test might be interesting although information from "initial tests" may be useful. Are the production and results linear with yields?

Line 164. Do you think these assumptions are important for the results.

Line 172: Sometimes the different properties of hygroscopicity is taken into account by a employing a bulk coefficient to the condensation term with an interval of [0 1]
This is not relevant here, or just too uncertain ?

Line 214. Refer back to section for readability

Line 220. Any reference to the entire model system ( van Noije et al ?) Given a reference which SOA parameterisation is used in that work

Line 225. Is the resolution the same as in the referenced version of the model.

Line 250. overpresented is likely not the right word

Line 325: Very low dry deposition. This should be the same for sulphate and elemental carbon.?

Line 329 -335. Unclear description, can be made shorter. Can be describe it as a combination of increase (production aloft) and decrease (aitken vs accumulation) ?

Line 346. Ocean production:  Any Ricco NPF above the boundary layer (outflow from continents?)

Line 354: No observation of NPF over ocean?

Line 359 Solitary "fig" word

Line 461 Increas → e

Section 3.4.2 Is the reduction in PM2.5 mainly due to reduction of SOA mass or are there other non-linear effects?

Line 501-506: Very unclear.

Line 514-520: Are the results from satellite and Aeronet consitent.  I do not expectant large analysis here but nice to point out.?

Line 524. The magnitude of AOD ?  → The increase of AOD varies by an order of magnitude of 0.003 to 0.03

---

## Author Comment (AC2)

We thank Anonymous Referee #1 for helpful suggestions and comments, which helped us improve the manuscript. Our point by point answers to the comments are presented below. Referee comments are in bold and our replies in body text.

**Referee #1 comments:**

***There are a number of figures and tables in the supplement which are never referred to in the paper. In addition, Fig 13 in the paper does not show the figure being described in the text.***

We will add references to supplementary figures and tables. The wrong Fig. 13 will be exchanged.

***In the following I detail changes/explanations that are needed:***

***Please explain how you calculate OH and O3. Are these read-in?***

No, the model has interactive chemistry. In the relevant parts of the atmosphere, OH and $O_3$ are calculated online. We have added the following to the Section 2.3.2:

"The $O_3$ and OH oxidant fields are calculated online in the chemistry code."

***Line 34: odd addition of numbers in "vegetation"***

Corrected.

***Line 46: please add references to models producing up to 121 Tg/yr SOA***

We have added the following to the text:

"IMAGES (Müller et al. 2009; Stavrakou et al 2009; Ceulemans et al. 2012) and IMPACT (Lin et al. 2012) models in Tsigaridis et al. (2014)"

***Line 47: While VBS schemes are "more intricate" they are essentially parameterizations. You need to state this.***

Changed "schemes" into "parameterisations" in the text.

***Line 52: "subsequently" is a strange word here. "In this theory"?***

Changed as suggested.

***Line 60: remove first "of"***

Done.

*Line 69: add Wang and Penner, 2009:*

*Wang, M. and J. E. Penner, 2009: Aerosol indirect forcing in a global model with particle nucleation, Atmos. Chem. Phys., 9, 239-260, www.atmos-chem-phys.net/9/239/2009/.*

*After all, this paper was published almost a full year before Merikanto et al., 2009*

Thank you for the suggestion. We have added the citation.

*Lines 101-103: You do not follow the mixing of soluble and insoluble particles? This seems highly unrealistic, since, for example SO4 would almost certainly condense onto insoluble particles. Are simulating sulfate condensation only on the soluble modes? If on both types of particles, how do you separate into the modes after condensation, since sulfate and dust forming on an Accumulation particle, could, after separation, form 2 Aitken mode particles or 1 Aitken mode and 1 accumulation mode, depending on the size of the original particle after condensation.*

We use the standard M7 formulation (Vignati et al. 2004), where sulfuric acid condenses onto soluble and insoluble particles. Ageing of insoluble particles takes place by both condensation of sulfuric acid and coagulation. The fate of particles formed by coagulation depends on the modes of the coagulating particles, as described by simple rules. Particles in the insoluble accumulation mode can only coagulate with soluble Aitken-mode particles, resulting in a transfer to the soluble accumulation mode. The sulfate condensation onto insoluble modes will lead to transfer of particles into soluble modes. The fraction of particles that will be transferred is calculated as the number of particles for which a layer of sulfate molecules with a thickness of one molecule can be created.

Vignati, E., Wilson, J., and Stier, P.: M7: An efficient size-resolved aerosol microphysics module for large-scale aerosol transport models, J. Geophys. Res., 109, D22 202, 10.1029/2003JD004485, 2004.

*Line 110: delete "more" in "A more detailed description", since no description is given.*

Done.

***Figure 1: Is there some reason that the reaction of isoprene and monoterpenes with NO3 is omitted?***

We started out to reproduce the method used by Jokinen et al. (2015), which only includes yields for reactions with OH and $O_3$. We had a discussion to include oxidation by $NO_3$ but due to time constraints of CMIP6 we were unable to implement this reaction.The chemistry does include $NO_3$ which will allow this reaction to be added in the future.

***From the figure the dotted line for the insoluble mode is only green for the Aitken model. So, it seems ELVOC and SVOC only condense on the insoluble Aitken mode. Why is this?***

That is because in M7 the insoluble accumulation and coarse modes only include a dust tracer.

***Line 139-140: does the diurnal emission variation for monoterpene at least depend on the local temperature variation? Sureely the model has this variable available. Please explain why it should not take this into account, if not and give the max/min dependence of the emission.***

As explained in the text, we normalize the daily mean emission to produce the emitted mass from an inventory. In order to use temperature we would need to simulate a full day first and then normalize the emission accordingly, which is not feasible to do.

 We have changed

"These variables are not available in the model and therefore" to:

 "Since we rely on emission inventories the daily emissions need to be normalized which makes the use of drought or temperature not feasible and therefore..."

***141: This reads like the monthly mean isoprene emission is 572.3 Tg/yr, but I know that is the annual emissions rate. I think you mean you employ a monthly varying emission with an annual mean of....***

We will rephrase this as follows:

"In this work we employ monthly varying isoprene and monoterpene emissions (with annual emissions of 572.3 Tg yr$^{-1}$ and 95.5 Tg yr$^{-1}$, respectively) from an inventory derived from MEGANv2.1..."

***175-176: Why wouldn't SVOC partition onto Accumulation and coarse modes if OA is present? (for example, by ELVOC condensation on these modes)?***

This was not clearly written. We have rephrased the text as follows:

"soluble Aitken, accumulation and coarse modes and insoluble Aitken mode."

***215: I gather "all other biogenic emissions" really just refers to isoprene? Why not say this?***

TM5 is a chemistry transport model and it also includes other biogenic VOCs in addition to isoprene and monoterpenes although these do not directly affect the aerosol phase.

***218-225: I think a paper should give sufficient information so that the reader does not need to read other papers in order to understand. Why not add a table with all annual emissions and references? After all, the emissions are one basic element affecting the evaluation***

We will add a table listing the annual emissions for aerosol and relevant precursors with their references.

***230: I don't think you mean SOA "on" the surface. Perhaps "in the boundary layer"? or in the first level of the model.***

Here we refer to the observations that we are using. We will rephrase to "in the boundary layer".

***246: The factor of 1.6 should be referenced, unless there is no reference. If basically unknown, then say so.***

This is documented in the paper by van Noije et al. (2021). We have included a reference to this paper.

van Noije, T., Bergman, T., Le Sager, P., O'Donnell, D., Makkonen, R., Gonçalves-Ageitos, M., Döscher, R., Fladrich, U., von Hardenberg, J., Keskinen, J.-P., Korhonen, H., Laakso, A., Myriokefalitakis, S., Ollinaho, P., Pérez García-Pando, C., Reerink, T., Schrödner, R., Wyser, K., and Yang, S.: EC-Earth3-AerChem, a global climate model with interactive aerosols and atmospheric chemistry participating in CMIP6, Geosci. Model Dev. Discuss. [preprint], https://doi.org/10.5194/gmd-2020-413, in review, 2020.

***247: do not use "on the surface"***

We will rephrase it to *"in the boundary layer".*

**250: change "overpresented" to "overrepresented"**

Done.

**313-314: here you first mention that there is no SOA formation from isoprene in OLDSOA. This should be stated earlier when talking about the sources from MEGAN (new and old).**

It is mentioned already on lines 144 and 214 that in OLDSOA SOA formation is based on monoterpene emissions from MEGANv1. We have added a sentence on both locations to point out that isoprene does not contribute to the formation of SOA.:

"Isoprene does not contribute to the SOA formation in the old scheme."

**326-327: can you speculate why the wet deposition fraction is higher in your model? Perhaps the vertical distribution places SOA higher in the atmosphere than in the other models?**

The fraction of wet deposition of SOA does change from 98.1% in OLDSOA to 98.9% in NEWSOA, but it is high in both cases. So it would seem that the wet deposition is somewhat increased due to additional production aloft. However, high wet deposition is also found in other models using the M7 aerosol module (Tsigaridis et al. 2014), which would indicate that it is an M7 module feature and not so much related to SOA, although the new formulation increases it further. Furthermore, many of the AeroCom models simulate SOA as emitted particles similarly to OLDSOA, which indicates that the difference from OLDSOA to AEROCOM models does not result from the vertical distribution.

**Figure 7: Are (c) and (d) concentrations in the surface layer?**

Yes, we will add this information in the caption accordingly.

**341-342: change to "using the additional particle formation" - I had to look back to see if you also include the Vehkamki scheme for sulfate in NEWSOA**

*Done.*

**352-357: please explain why southern ocean nucleation mode particle numbers are higher in OLDSOA than NEWSOA. After all, both these schemes include binary homogeneous nucleation**

The two simulations handle the newly produced particles differently; in NEWSOA nucleation mode particles growth to 5nm is parameterised based on sulfuric acid and/or

ELVOC while in OLDSOA no growth is assumed. We have added an explanation into the text:

"Furthermore, OLDSOA shows higher concentrations (and NPF) over Antarctica. The reason for this is the different handling of nucleation mode particles in the two simulations; in NEWSOA the initial particle growth to 5 nm in diameter is parameterized while in OLDSOA no growth of particles is assumed (see Sect. 2.3.5 for details)."

**361: missing e in "increase"**

Corrected.

**365: "of of"**

Corrected.

**369-370: I do not understand why particle formation in NEWSOA isn't always larger than in OLDSOA, since the former includes RiCCO AND binary sulfate formation**

In NEWSOA the new particle formation is calculated at 5 nm while in OLDSOA the new particle formation from Vehkamäki et al. (2002) is used as is. That parameterisation can produce a high number of particles with diameters around 1 nm.

**371-373: what is meant by "compensated"? this explanation needs some expansion – it is hard to follow.**

Due to the suggestion from another reviewer we have looked into it more closely and both stations with higher concentrations in OLDSOA are located in Antarctica. We will rephrase the last paragraph as follows::

"In general the updated SOA and new NPF parameterisation caused an increase in particle number concentrations, but at two Antarctic stations the concentrations decreased. Main reason is the change in NPF parameterisation where in OLDSOA the NPF production is calculated directly by BHN nucleation and in NEWSOA the early growth to 5 nm diameter particles is parameterised, which are therefore less numerous."

**384: I think you do not mean observed N2O, since this is a gas. Also, it appears CCN particles do not reach 5000/cm3: the scale only goes to 3000/cm3, but Vavill goes to 5000/cm3. BTW if you are referring to the top 2 lines of Fig 9b, there is no scale, so one cannot tell if the observation goes to 5000/cm3.**

This was incorrectly written as we were using both N20 and CN (condensation nuclei) interchangeably. However, it is not N2O but N20 and refers to particles with diameters larger than 20 nm which is noted as CN in the figure. We will change the notation in the figure.

And for the particle concentrations, in the text we refer to CN, which reaches almost 5000 cm$^{-3}$ at Birkenes, Norway (middle panel of Fig 9b, left (red) axis). The figures include both CN and CCN using different color lines, here CN comprises the particles larger than 20 nm (or 3 nm in Vavihill) in diameter and CCN are the particles larger than 70 nm in diameter. In the line plots of the figure the right hand side axis refers to CCN and left hand axis to CN. This is noted by the matching colors on "CN" and "CCN" in the axis label. We will change these to $N_{20}$ and $N_{100}$ for clarity. As requested we will add a colorbar for the upper panels. We will also revise the main text for more clarity and add text in the caption to describe the figure properly.

***Figure 11: in (e) and (f), I assume the 2 NMB's and RMSE's are for PM2.5 and PM10, but they are not labeled or in the caption. Also, what are the 2 numbers after R: in the upper 2 panels?***

We have removed these since they are not referred to in the text.

***Figure 13: the label on the graph states that both a and b are the difference between NEWSOA and OLDSOA, whereas the caption states the correct difference.***

This was an error in the production stage: somehow another figure was exchanged with the actual paper figure. We will insert the correct figure here (shown below).

[Figure]

***472: Where is OLDSOA compared with MODIS and AATSR? The figure caption of figure 13 states that this figure is only NEWSOA. Also, it appears that both panels in fig 13 have most of the tropics and subtropics overestimated, so how do you estimate an underestimated AOD? Much of the paragraph after this sentence in line 472 cannot be understood based on fig. 13.***

See answer to comment above.

*So, when you finally reference the supplemental figures, I see that the supplement Fig S10 and S11 compares OLDSOA and NEWSOA with MODIS and AATSR. I find it strange that this is perhaps the first supplemental figure (or table) that is meant to be referenced. Why have figures and tables in a supplement if they are not referenced?*

*BTW, I would at least include an annual average difference of OLDSOA and NEWSOA with MODIS and AATSR in the main text. As it stands, the main result of the data shown in the current Fig 13 was already discussed.*

As stated earlier we will insert the correct figure in the main text and move the current figure to the supplement as noted earlier. The annual means have been noted in the text on lines 473 (MODIS) and 491 (AATSR).

*552: add a comma after "events"*

Done.

*553: explain what the "numerical issues" are. This is the first mention of this fact. Also, I would add "perhaps" since you do not actually analyze the reason for low CCN sized particles (by comparison with observations, for example)*

It is mentioned already in Section 3.3.1 on particle concentrations at selected sites on line 402, but to be clear we have added there some explicit issues:

"(e.g. errors from operator splitting, unrealistic size distributions due to mode merging)"

*567: The change in AOD is ultimately from the change in particle number, but must actually be due to the change in size distribution, I think. This sentence should be expanded to remind the reader of the processes leading to the change in AOD*

Thank you for a good suggestion. We will rephrase with the following:

"Therefore, the change in AOD results mainly from the different shape of the size distribution. Because the condensation of SOA in NEWSOA is physically described the particle concentrations in the optically important size range are higher than in OLDSOA."

---

## Author Comment (AC3)

We thank Anonymous Referee #2 for helpful suggestions and comments, which helped us improve the manuscript. Our point by point answers to the comments are presented below. Referee comments are in bold and our replies in body text.

**Referee #2 comments:**

**General comments:**

*The TM5-MP model description lacks the spatio-temporal model resolution used for this study. In the evaluation and conclusions, the time resolution becomes clear. In section 2.4 the simulation period, and later in the text also the horizontal resolution, are mentioned. For the reader it would be easier to get the full information on spatial and temporal resolution, such as the time period for the simulations either right at the beginning in Section 2.1 or in Section 2.4 in a more coherent way.*

We will collate the simulation details in Section 2.4 for more clarity.

*In Section 2.3 the earlier version of TM5 is described and it is stated that it uses biogenic monoterpene emissions from MEGAN v1 and does not include isoprene emissions for SOA formation. In contrast, the new version calculates SOA from isoprene and monoterpenes, based on MEGAN v2 inventories. In section 2.4 the authors describe the simulations to be compared, one simulation using the older version "OLDSOA" and a second simulation for the same time period using the new formation schemes "NEWSOA". The difference between these two simulation configurations are not only the novel SOA and new particle formation schemes, but also the changed biogenic emissions plus, in the old scheme only monoterpene-derived SOA was included. For disentangling the contributions from the updated biogenic emissions from the contributions by the extended SOA treatment, I would suggest to add sensitivity simulations. I wonder how does the NEWSOA scheme perform compared to OLDSOA while using the same MEGAN v1 emissions? Also, how do NEWSOA and OLDSOA compare when both using MEGAN v2 emissions, even if OLDSOA does not form isoprene-derived SOA? I would not expect the difference in emissions shown in Figure 2 apply in a linear way to SOA formation. In Section 3.1 the last paragraph points into the direction. Additional sensitivity simulations with both emission inventory versions would help to understand what difference is caused by the changed emissions and what difference is related to the novel SOA formation pathways.*

This is a good suggestion and we have performed an additional sensitivity simulation similar to the OLDSOA simulation but using surrogate SOA emissions with the MEGAN v2 monoterpene emissions. The main results will be discussed in the paper.

***In section 2.3.2 the production of ELVOC and SVOC is described. In the introduction, the authors do mention low volatility compounds (LVOC), but do not explicitly state here why they assume two products of the isoprene and monoterpene reactions (ELVOC and SVOC) and leave out the possible LVOC products. It would be helpful for the reader to know the reason for this decision.***

We apply a hybrid method that can consider kinetic condensation to surface area but also consider products that reach equilibrium with gas and particle phases (e.g. Riipinen et al., 2011). We lump the products into two different volatility classes. ELVOC represents the fraction that has such a low volatility it can participate in the growth of the nanometer scale particles (e.g.Tröstl el al. 2016).  The other SOA precursors in our mechanisms are assumed to represent those that reach equilibrium with the gas phase within one time step. These are partitioned to the aerosol phase according to mode OA mass. Even though SOA cannot re-evaporate this represents the semi-volatile fraction of the condensable VOC. In essence, the LVOC is lumped into the same fraction as SVOC in our model.

We have extended the text in the introduction:

"...are often separated into lumped species. In the present model we separate low-volatility products into semi-volatile VOC (SVOC) and extremely low-volatility VOC (ELVOC). " .

Furthermore, we have added following to Sect.  2.3.2:

"In this work we assume that SVOC does not re-evaporate."

And in Section 2.3.3 describing the partitioning to particles:

"The equilibrium model is assumed to be irreversible, since the yields are determined in the equilibrium state."

Riipinen, I., Pierce, J. R., Yli-Juuti, T., Nieminen, T., Häkkinen, S., Ehn, M., Junninen, H., Lehtipalo, K., Petäjä, T., Slowik, J., Chang, R., Shantz, N. C., Abbatt, J., Leaitch, W. R., Kerminen, V.-M., Worsnop, D. R., Pandis, S. N., Donahue, N. M., and Kulmala, M.: Organic condensation: a vital link connecting aerosol formation to cloud condensation nuclei (CCN) concentrations, Atmos. Chem. Phys., 11, 3865–3878, https://doi.org/10.5194/acp-11-3865-2011, 2011.

Tröstl, J., Chuang, W., Gordon, H. *et al.* The role of low-volatility organic compounds in initial particle growth in the atmosphere. *Nature* 533, 527–531 (2016). https://doi.org/10.1038/nature18271

***Table 2 nicely shows the SOA budget together with some literature values. In this study, the authors differentiate between monoterpene-derived SOA and isoprene-derived SOA, a distinction which cannot be found in the literature cited in Table 2. A study focusing on isoprene derived SOA could add some more information on how to interpret the values for isoprene-derived SOA. Please consider comparing to the global isoprene-derived SOA values in Table 4 given in Stadtler et al. 2018 (https://doi.org/10.5194/gmd-11-3235-2018). I did not find a comparable study for monoterpene- derived SOA on a global scale, but it might also exist.***

Just as the reviewer, most of the studies we have found only report the total SOA production without information on the production pathway. However, we will add the data from Stadtler et al. (2018) and a discussion in the paper.

***In Section 3.3 the authors start to compare the OLDSOA and NEWSOA simulations to observed particle number concentrations (Figure 8). Although this kind of scatter plot is a common evaluation method, the comparison shown here does not include any error or uncertainty indication, either for the observations, nor for the statistical error caused by comparing grid cells to point measurements. Swall et al. 2009 (https://doi.org/10.1016/j.atmosenv.2008.10.057) describe and discuss the incommensurability problem arising from comparison of point measurements to grid cell values. According to Swall et al. even if a perfectly performing model and measurements without observational error, the scatterplot will not show modeled and observed values on a one-to-one line. The same holds true for Figure 14 and comparison to AERONET stations, Figure 10 in comparison to IMPROVE stations and Figure 11 in comparison to EMEP stations. The authors should consider including information on uncertainty derived from the comparison approach and, if available, the observational errors. A model error would be nice-to-have, but I dare not to ask for that.***

We agree that there are several sources of errors as also stated in the studies by Schutgens et al. (2016a, 2016b and 2017). To limit the errors from temporal sampling we have done colocation of the model data with the observations at the hourly level. In Schutgens et al. (2016a) it is estimated that constructing yearly averages from daily sampling will produce errors of 7–17%. However, we use hourly sampling (when possible, e.g. EMEP can have daily or monthly data) which should lead to smaller sampling errors. In Schutgens et al. (2016b) they show errors of at least 30% and up to

80% in simulated black carbon concentrations compared to in-situ observations. Furthermore, in monthly data they note errors to be typically between 10%–40%. They also point out that one should do spatio-temporal collocation which we have done, but still some error will remain. We have done this to limit the errors as much as possible.

We will add error bars into the figures in question using standard errors for the means. Furthermore, we will add a discussion on the error in Sect. 2.5 with a description of the data used in evaluation where we also note that we are doing colocation of observations and model data.

Schutgens, N. A. J., Partridge, D. G., and Stier, P.: The importance of temporal collocation for the evaluation of aerosol models with observations, Atmos. Chem. Phys., 16, 1065–1079, https://doi.org/10.5194/acp-16-1065-2016, 2016 ab.

Schutgens, N. A. J., Gryspeerdt, E., Weigum, N., Tsyro, S., Goto, D., Schulz, M., and Stier, P.: Will a perfect model agree with perfect observations? The impact of spatial sampling, Atmospheric Chemistry and Physics, 16, 6335–6353, https://doi.org/10.5194/acp-16-6335- 2016, https://www.atmos-chem-phys.net/16/6335/2016/, 2016 b.

Schutgens, N., Tsyro, S., Gryspeerdt, E., Goto, D., Weigum, N., Schulz, M., and Stier, P.: On the spatio-temporal representativeness of observations, Atmos. Chem. Phys., 17, 9761–9780, https://doi.org/10.5194/acp-17-9761-2017, 2017.

*In whole Section 3 the description of the results and comparison is blended with the discussion of the model errors and limitations. The authors should consider editing the subsections in Section 3 such that the description of the results is separated from the discussion and interpretation of the model strengths and weaknesses. A clear strength of this study is the evaluation with a variety of observations, but it makes the manuscript difficult to read if the interpretation and discussion is mixed into the comparisons to the old model and observations. The authors should consider to collect all the discussion points of NEWSOA schemes' limitations and advantages and to write them into a single "Discussion" Section in a concise way.*

Thank you for the suggestion. We will do so.

*Out of curiosity, why was it not possible to include isoprene and monoterpene oxidation by NO3 in this study? As you mention, it limits the current model performance, but should be considered in a future study. I wonder what reason was prohibiting to implement this additional oxidation reaction in the NEWSOA scheme.*

We started out to reproduce the method used by Jokinen et al. (2015), which only includes yields for reactions with OH and $O_3$. It is true that it lacks the $NO_3$ oxidation. We had a discussion to include oxidation by $NO_3$ but due to time constraints of CMIP6 we were unable to implement this reaction at this time. As noted in the text, the chemistry does include $NO_3$ which will allow this reaction to be added in the future as we have noted in the text.

***Technical corrections:***

***In the abstract line 11 – 14 it is written "Compared to the old scheme, the new schemeis increasing the number concentrations across the observation stations while still underestimating the observations. The total aerosol mass concentrations in the US show a much better seasonal cycle and removal of a clear overestimation of concentrations.", which is confusing the reader. First the authors speak of "still underestimating" but the next sentence reads "removal of overestimation". Please make it clearer here.***

These two things are not exactly the same. The first sentence refers to number concentrations and the latter to mass concentrations. Nevertheless, it is a bit unclear so we have changed the latter sentence to:

"The organic aerosol mass concentrations in the US show a much better seasonal cycle and no clear overestimation of mass concentrations anymore."

***In Line 34 there is some number in the word "vegetation" "vegetat589527ion". Line 431 and 432: Two sentences in a row starting with "However".***

We will correct these.

*I*n *Figure 13 the plot labels indicate [NEWSOA-OLDSOA], but according to the caption of Figure 13 the annual mean difference of NEWSOA and MODIS (a), NEWSOA and AATSR (b) are shown. Please change the label of the plots.*

This will be corrected. It was an error introduced during the production of the preprint version, where we mixed another figure with the article figure.

***Line 504 it says "However, compared to OLDSOA the absolute bias across all stations increases in NEWSOA by 0.007 to a low bias of 0.003", should the verb be "decreases"?***

We will correct this as suggested.

---

## Author Comment (AC4)

We thank Anonymous Referee #3 for helpful suggestions and comments, which helped us improve the manuscript. Our point by point answers to the comments are presented below. Referee comments are in bold and our replies in body text.

*Referee #3 comments and suggestions:*

*While none of the parameterizations are new, the more physical approach is a useful addition to aerosol modelling tools, in particular for M7 and derivatives thereof. The article is in general well structured although sometimes it is a bit unclear when the authors describe the results from the old or the new version. A somewhat more problematic issue is that the authors often conclude that this is caused by emission changes and this is due to secondary aerosol formation. While it may be obvious for most cases I think the authors should do a sensitivity test that only show the impact of the emission changes. The assumptions on yields is another parameter that should be discussed although the emission test will probably work as a first approximation on understanding yields.*

To analyse the effect of emissions on SOA production we have performed an additional sensitivity simulation similar to the OLDSOA simulation but using surrogate SOA emissions with the MEGAN v2 monoterpene emissions that are used for the NEWSOA simulation. The main results will be discussed in the paper.

Sporre et al. (2020) analyzed the behavior of the model with our NEWSOA scheme using a set of sensitivity experiments. By scaling the SVOC and ELVOC yields up and down by 50%, the resulting SOA mass was increased / decreased by almost 50% (Sporre et al., 2020, Fig. 6). Products from isoprene and monoterpenes contributed about 80% and 20% to the SOA mass (Sporre et al., 2020, Fig. 7). The response of the number concentration and number size distributions was rather non-linear since it very much depends on the competition between new particle formation and growth of existing particles. The balance between the two processes was changed by adapting the SVOC and ELVOC yields.

We will add the text above also to Sect. 2.3.2 the manuscript.

Sporre, M. K., Blichner, S. M., Schrödner, R., Karset, I. H. H., Berntsen, T. K., van Noije, T., Bergman, T., O'Donnell, D., and Makkonen, R.: Large difference in aerosol radiative effects from BVOC-SOA treatment in three Earth system models, Atmos. Chem. Phys., 20, 8953–8973, https://doi.org/10.5194/acp-20-8953-2020, 2020.

*It may beyond the scope of the article and or due to missing capabilities of the model I am wondering if the authors also considered comparing the Ångstrøm component in particular in light of the focus of aerosol size changes from the new parameterisaton.*

We find this an interesting idea, but it is beyond the scope of this paper.

*Minor / technical issues.*

*The introduction splits the organic compounds between ELVOC, LVOC and SVOC, Later on it only use ELVOC and SVOC. My usual interpretation is that SVOC s reserved for thos compounds that can commonly re-evaporate. As I understand none of the VOCs described in this work is re- evaporating. I think this should be stated explicitly if the authors continue to use SVOC.*

That is correct, we assume that all of the SVOC from the oxidation will reach equilibrium within the time step of the model. We will add the following sentence in Section 2.3.2:

"In this work we assume that SVOC does not re-evaporate."

And in Section 2.3.3 describing the partitioning to particles:

"The equilibrium model is assumed to be irreversible, since the yields are determined in the equilibrium state."

*Abstract line 11. "clearly lower as is" new versus old ?*

This is indeed a bit unclear. Here we compare to the observations and we have rephrased the sentence as:

"However, the modelled concentrations of formed particles are clearly lower than in observations as is the subsequent growth to larger sizes."

*Line 32: Missing in first volatility word*

Corrected.

*Line 34. Numbers inside "vegetation"*

Corrected

*Line 39: Two "rathers" are rather too much. Can one of them e.g. the first be replaced by a number / fractional estimate.*

Indeed, we have rephrased the sentence; it now reads:

"Emissions of BVOCs constitute roughly 90% of total VOC emissions, but due to complex chemistry the actual processes and the total amount of SOA formation is rather uncertain (Tsigaridis et al. 2014)."

*Line 76: "between" → "divided between" ?*

Corrected as suggested.

*Line 93, but this has also consequences for the remaining manuscript. Does "trace gases "include OH? If not the description of chemistry must include information on the OH concentration used in the model.*

*If OH is part of the chemistry. Will the SOA chemistry have noticeable impact on the OH concentration?*

Both simulations have the same emissions and chemistry of monoterpene and isoprene despite the fact that the surrogate SOA emissions are consistent with monoterpenes from MEGANv1. Chemical reactions with these VOCs and OH and $O_3$ are already present in OLDSOA but describe only the removal of VOCs in OLDSOA while SOA production is provided separately. In NEWSOA using the chemical reactions describing the removal of isoprene and monoterpenes also the production of the SOA mass is calculated. The change in aerosol fields will cause some change in gas-phase concentrations of oxidants, but there is no direct influence in the reactions themselves. We have explored how big the impact is, and the change in annual global mean of OH concentrations in all model levels is well below 1%. At the surface there are regions with annual mean change of 1–2% while higher up it changes to 5% at maximum. In our view the change is relatively small.

*Figure 1: No evaporation of SVOC?*

As in Jokinen et al. (2015) we assumed that the particles are in equilibrium with the gas phase and the yields are calculated to take into account only the mass remaining in the particles.

*Line 121. Are isoprene and monoterpene emitted at ground level or follow the height distribution of the oldSOA.*

They are emitted at ground level. We have added the following on line 143 (after declaring the emission strength and reference to Fig. 2):

"It should be noted that the gases are emitted at ground level."

*Line 143: Unclear. "geographical distribution ..." As far as I understand the model reads regridded monthly mean emissions from a specific MEGAN set-up. TM5 does not do anything to the emissions themselves e.g. related to meteorological fields? so it is just how the emission look like from the model (except for the diurnal cycle)*

That is correct. We have changed this to "; the geographical distribution of emissions is shown in Fig. 2a,c"

*Line 156. Choice of yield looks a bit arbitrary. Tuning ? A sensitivity test might be interesting although information from "initial tests" may be useful. Are the production and results linear with yields?*

Yes, SOA mass in close approximation scales as the yields. Sporre et al. (2020) analyzed the behavior of the model with our NEWSOA scheme using a set of sensitivity experiments. By scaling the SVOC and ELVOC yields up and down by 50%, the resulting SOA mass was increased / decreased by almost 50% (Sporre et al., 2020, Fig. 6).

Regarding the choice of yields, for monoterpene the yields are directly from Jokinen et al. (2015) but for isoprene we have modified the total yield from 5% to 1% because SOA production from isoprene in our initial tests was very high. Furthermore, as explained in the text Kroll et al. (2005) report mass yields of 0.9%-3.3% which align with our choice of 1% molar yield. We added in the caption a citation to Jokinen et al. (2015) and note to check the text for explanation on isoprene yields.

Sporre, M. K., Blichner, S. M., Schrödner, R., Karset, I. H. H., Berntsen, T. K., van Noije, T., Bergman, T., O'Donnell, D., and Makkonen, R.: Large difference in aerosol radiative effects from BVOC-SOA treatment in three Earth system models, Atmos. Chem. Phys., 20, 8953–8973, https://doi.org/10.5194/acp-20-8953-2020, 2020.

*Line 164. Do you think these assumptions are important for the results.*

We think that when NEWSOA is compared to OLDSOA, where SOA is put in one mode as mass only, the assumptions are important in understanding the differences.

*Line 172: Sometimes the different properties of hygroscopicity is taken into account by a employing a bulk coefficient to the condensation term with an interval of [0 1]*

*This is not relevant here, or just too uncertain ?*

We assume that the applied yields give the amount of SOA which will remain on the particles following Jokinen et al. (2015). Therefore, all of the SOA mass is distributed to the particles by organic mass or by surface area of the existing particles and condensation is not solved explicitly. So in our view this is not relevant here.

Jokinen, T., Berndt, T., Makkonen, R., Kerminen, V.-M., Junninen, H., Paasonen, P., Stratmann, F., Herrmann, H., Guenther, A. B., Worsnop, D. R., Kulmala, M., Ehn, M., and Sipilä, M.: Production of extremely low volatile organic compounds from biogenic emissions: Measured yields and atmospheric implications, Proceedings of the National Academy of Sciences, 112, 7123–7128, https://doi.org/10.1073/pnas.1423977112, http://www.pnas.org/content/112/23/7123.abstract, 2015.

**Line 214. Refer back to section for readability**

We do not think this will improve the readability since it refers to all of this section until this point.

**Line 220. Any reference to the entire model system ( van Noije et al ?) Given a reference which SOA parameterisation is used in that work**

The EC-Earth3-AerChem version is documented in van Noije et al. (2021), and uses the new SOA and NPF parameterisation as explained there.

van Noije, T., Bergman, T., Le Sager, P., O'Donnell, D., Makkonen, R., Gonçalves-Ageitos, M., Döscher, R., Fladrich, U., von Hardenberg, J., Keskinen, J.-P., Korhonen, H., Laakso, A., Myriokefalitakis, S., Ollinaho, P., Pérez García-Pando, C., Reerink, T., Schrödner, R., Wyser, K., and Yang, S.: EC-Earth3-AerChem, a global climate model with interactive aerosols and atmospheric chemistry participating in CMIP6, Geosci. Model Dev. Discuss. [preprint], https://doi.org/10.5194/gmd-2020-413, in review, 2020.

**Line 225. Is the resolution the same as in the referenced version of the model.**

TM5-MP can be run in 1x1 degree and 3x2 degree resolutions. In EC-Earth the atmospheric component IFS is running on T255L91 grid and is coupled with TM5-MP running at 3x2 degree resolution. This is the same as in the model version documented in van Noije et al. (2021).

van Noije, T., Bergman, T., Le Sager, P., O'Donnell, D., Makkonen, R., Gonçalves-Ageitos, M., Döscher, R., Fladrich, U., von Hardenberg, J., Keskinen, J.-P., Korhonen, H., Laakso, A., Myriokefalitakis, S., Ollinaho, P., Pérez García-Pando, C.,

Reerink, T., Schrödner, R., Wyser, K., and Yang, S.: EC-Earth3-AerChem, a global climate model with interactive aerosols and atmospheric chemistry participating in CMIP6, Geosci. Model Dev. Discuss. [preprint], https://doi.org/10.5194/gmd-2020-413, in review, 2020.

**Line 250. overpresented is likely not the right word**

We have changed this to overrepresented.

**Line 325: Very low dry deposition. This should be the same for sulphate and elemental carbon.?**

The fraction of particle removal by dry deposition varies between 1.1% to 2.8 % for BC (2.2%), OA (1.7%), SOA (1.1%), and SO4 (2.8%). It is shown in Section 4.1.4 of Tsigaridis et al. (2014) that models that employ M7 show very low dry deposition rates for organic aerosol. Furthermore, the SOA dry deposition fraction varies between 1.1% in the NEWSOA and 1.9% in the OLDSOA simulation, which seems to suggest that production aloft in NEWSOA increases the importance of wet deposition. For other compounds there is almost no change.

**Line 329 -335. Unclear description, can be made shorter. Can be describe it as a combination of increase (production aloft) and decrease (aitken vs accumulation) ?**

We will rephrase this:

"In NEWSOA the SOA mass is distributed into the aerosol size distribution more evenly than in OLDSOA. This means that mass in accumulation mode is decreased while Aitken mode mass is increased which could suggest lower lifetime (see Schutgens and Stier (2014) for modewise lifetimes in M7). However, in NEWSOA the SOA production aloft is clearly higher meaning that the deposition takes more time leading to a longer lifetime. "

Schutgens, N. A. J. and Stier, P.: A pathway analysis of global aerosol processes, Atmospheric Chemistry and Physics, 14, 11 657–11 686, https://doi.org/10.5194/acp-14-11657-2014, https://www.atmos-chem-phys.net/14/11657/2014/, 2014.

**Line 346. Ocean production: Any Ricco NPF above the boundary layer (outflow from continents?)**

There is some production over oceans but it is very low. This is mainly because the model does not include any monoterpene or isoprene emissions from the oceans and all of the NPF over oceans depends on the continental outflow of these gases. Marine monoterpene and isoprene emissions are low and uncertain as noted in the manuscript.

**Line 354: No observation of NPF over ocean?**

We did not use any observations over oceans. We have made it clearer by adding "modelled" before NPF rate.

**Line 359 Solitary "fig" word**

Removed.

**Line 461 Increas → e**

Corrected.

**Section 3.4.2 Is the reduction in PM2.5 mainly due to reduction of SOA mass or are there other non- linear effects?**

Table 3 seems to be a bit unclear, but it lists only the OA concentrations as stated in the caption, we will clarify the table in the revised manuscript. Similarly, in the main text we only discuss organic mass in PM2.5 not the total PM2.5. Nevertheless, aerosol dynamics are inherently non-linear, and here the scheme in OLDSOA is unphysical since it distributes all of the produced SOA mass into Aitken mode in the model. This will cause the Aitken mode to grow rapidly into the accumulation mode with longer lifetimes, while in NEWSOA the mass is more evenly distributed with less mass in the accumulation mode. This is more prominent for the IMPROVE stations, but it will also affect the organic mass in PM2.5 in Europe.

**Line 501-506: Very unclear.**

We will rephrase this as follows (note that due to another reviewer's suggestion the seasonal cycle will be shown separately for both hemispheres):

"Figure 14 shows the comparison of AOD measured by AERONET and modelled AOD in NEWSOA (a), and OLDSOA (b). Fig. 14c shows the hemispheric seasonal cycles of AOD in AERONET observations, in NEWSOA and OLDSOA. Additionally, Fig. 14d presents a map showing the station locations and their regional groupings for the statistics in Table 4.

In general the simulations show similar behaviour where they overestimate the low AOD and underestimate the high AOD observations. However, in NEWSOA the absolute bias across all stations improves in NEWSOA to 0.184 with a low bias of 0.003 from 0.176 in OLDSOA. Furthermore, the correlation coefficient (R; see last row of Table 4 for the values across stations) increases by 0.009. Visually both simulations show a very similar deviation from the observations."

**Line 514-520: Are the results from satellite and Aeronet consitent. I do not expectant large analysis here but nice to point out.?**

The AERONET map in Fig. 15b and satellite map figures — the manuscript included a wrong Fig. 13, the correct one is included below and will be changed in the paper — and the correct satellite comparison maps show similar changes. In the Amazon both satellites show a decrease in AOD as do the AERONET sites. Europe, the US, and Southeast Asia show similar behaviour for both satellite instruments and AERONET. For Australia, the AERONET comparison shows a similar behavior as the MODIS instrument. Summarizing, the AERONET comparison is consistent in the direction of the change with satellite instruments in most regions.

We will add a discussion along these lines to the paper:

"A comparison of biases in AERONET (Fig. 15b) and satellite retrievals (Fig. 13) —note that the color scales do not match — shows qualitative agreements between the two sets of observations in most locations and times . Over the Amazon and Southeast Asia both satellites show a decrease in AOD as do the AERONET sites. Over Europe and the US both satellite instruments and AERONET show small or no change. It is noteworthy that in Australia where satellite retrievals do not agree, the AERONET comparison shows a similar behavior with the MODIS instrument."

[Figure]

*Line 524. The magnitude of AOD ? → The increase of AOD varies by an order of magnitude of 0.003 to 0.03*

Changed as suggested.

---

## Author Comment (AC5)

We thank Anonymous Referee #4 for helpful suggestions and comments, which helped us improve the manuscript. Our point by point answers to the comments are presented below. Referee comments are in bold-italic and our replies in body text.

*Referee #4 comments and suggestions:*

*General points:*
*Throughout, the reader would benefit from some more clarity or consistency as to how the two schemes are referred. The terms "NEWSOA" and "OLDSOA" could be introduced earlier to help the reader keep track of which parts of the model description relate to which results. For example, referring to the "TM5 aerosol distribution" (as in the Fig 9 caption); it is not clear whether this is TM5 with the SOA developments included or not.*

We will add a sentence stating the simulation names already in Section 2.3 when describing the schemes. Furthermore, we will correct the Fig. 9 caption and Sect. 3.3.1 to show that the simulation is NEWSOA.

*Similarly, it would be clearer throughout to refer to the model performance being improved or degraded relative to observations rather than increased or decreased, as there are a lot of numerical values that increase and decrease.*

We will check the text and change to improve/degrade when appropriate.

*Minor / specific points:*
*Line 10-11: Reword to clarify which value (modelled or observed) is lower than which.*

We have clarified this by adding "modelled".

*Line 17-18: Reword to clarify what the retrievals or observations are of (AOD?)*

"of AOD" added

*Line 34: correct "vegetation"*

Corrected

*Line 150: this section could be clarified to confirm whether the split of the 15% and 5% yields*

*into ELVOCs and SVOCs (I.e., 1% ELVOC and 14% SVOC for the monoterpene + OH reaction) is also based on Jokinen et al., (2015) - if not how were those values determined? If the total values and the split values are from Jokinen et al., (2015) then this citation should be added to the Table 1 caption.*

For monoterpene the yields are from Jokinen et al. (2015) but for isoprene we have modified the total yield from 5% to 1% because SOA production from isoprene in our initial tests was very high. Furthermore, as explained in the text Kroll et al. (2005) report mass yields of 0.9%-3.3% which align with our choice of 1% molar yield. We added in the caption a citation to Jokinen et al. (2015) and note to check the text for explanation on isoprene yields.

Jokinen, T., Berndt, T., Makkonen, R., Kerminen, V.-M., Junninen, H., Paasonen, P., Stratmann, F., Herrmann, H., Guenther, A. B., Worsnop, D. R., Kulmala, M., Ehn, M., and Sipilä, M.: Production of extremely low volatile organic compounds from biogenic emissions: Measured yields and atmospheric implications, Proceedings of the National Academy of Sciences, 112, 7123–7128, https://doi.org/10.1073/pnas.1423977112, http://www.pnas.org/content/112/23/7123.abstract, 2015.

Kroll, J. H., Ng, N. L., Murphy, S. M., Flagan, R. C., and Seinfeld, J. H.: Secondary organic aerosol formation from isoprene photooxida- tion under high-NOx conditions, Geophysical Research Letters, 32, https://doi.org/10.1029/2005GL023637, https://agupubs.onlinelibrary. wiley.com/doi/abs/10.1029/2005GL023637, 2005.

*Line 164: can you clarify in this section whether the equilibrium partitioning approach is reversible or irreversible?*

The equilibrium model is irreversible, because we assume yields and therefore the produced SOA mass to represent the amount of SOA remaining in the particles in equilibrium state. Furthermore, we assume that equilibrium is reached within the model time step. To clarify this we have added the following to the text:

"The equilibrium model is assumed to be irreversible, since the yields are determined in the equilibrium state."

*Line 175-176: this implies that ELVOCs cannot condense onto particles in the nucleation mode but could you add a note to confirm? If so, does that leave a gap - i.e., ELVOCs can grow particles up to 5 nm diameter, but not beyond?*

This was incorrectly written. We have changed the indices so that i refers to ELVOC condensation and j to SVOC condensation. So i stands for all soluble modes and

insoluble Aitken mode as has already been explained in the text earlier. And j stands for Aitken, accumulation and coarse soluble modes as well as insoluble Aitken mode.

**Line 191-192: In Riccobono et al., (2014), BioOxOrg represented the product of oxidation of monoterpenes by OH specifically so this description could be clarified slightly to explain that you are including a wider set of products (by using ELVOCs).**

Agreed. The following sentence was added: „Whereas BioOxOrg represents the products from oxidation of monoterpenes by OH, our ELVOC includes a wider set of oxidation products."

**Line 218+: Add a note here to clarify whether / if all the other emission sources are identical between your simulations (i.e., the only difference is OLDSOA -> NEWSOA).**

Only the SOA differs between the simulations: in OLDSOA it is read in from a file and in NEWSOA it is calculated online. Both simulations have the same emissions and chemistry of monoterpene and isoprene despite the fact that the surrogate SOA emissions in OLDSOA are consistent with monoterpenes from MEGANv1. Chemical reactions with these VOCs and OH and $O_3$ are already present in OLDSOA but describe only the removal of VOCs while SOA production is provided as an external file. In NEWSOA using the chemical reactions describing the removal of isoprene and monoterpenes also the production of the SOA mass is calculated.

We will add a note that the emissions are the same in both simulations. As requested by another reviewer we will also add a table showing the emissions and corresponding references.

**Line 250: should "overpresented" be "overrepresented"?**

Corrected

**Line 358: stray "fig" here**

Corrected

**Line 360+: it would be useful to restate in this section (and the Figure 8 caption) where these stations are, even just by reference to Table S3.**

We have added a reference to the table in the first sentence:
"Figure 8 shows the annual means from the NEWSOA and OLDSOA simulations compared to the observed particle number concentrations at the stations listed in Table S3."

*Line 361: correct "increas"*

Corrected

*Line 368-369: can you specify at which two stations the particle number concentration decreased in NEWSOA relative to OLDSOA? This may aid with the subsequent explanation.*

Good suggestion, and we found out that these are two stations in Antarctica "South Pole" and "Neumayer". Our explanation for the reasons will be revised as follows:

"In general the updated SOA and new NPF parameterisation caused an increase in particle number concentrations, but at two Antarctic stations the concentrations decreased. Main reason is the change in NPF parameterisation where in OLDSOA the NPF production is calculated directly by BHN nucleation and in NEWSOA the early growth to 5 nm diameter particles is parameterised (see Sect. 2.3.5.), which are therefore less numerous."

*Line 479: I couldn't find a Fig. S13*

We made an error in producing the uploaded files and this figure was actually inserted in place of Fig. 13. We will add this figure in the supplement and put the correct Fig. 13 in the paper as noted in the next point.

*Line 490+: this description, in terms of the differences between the MODIS and AATSR comparisons, doesn't seem to align with Figure 13 – can you check that this is the correct Figure? (this may relate to below comment about Figure 13 caption/legend)*

Indeed when finalizing the upload version the figures in the supplement and manuscript were not correct. We have now put here the proper figure (see below) and the figure that was incorrectly placed has been moved to the supplement.

[Figure]

a) AOD difference [MODIS]

b) AOD difference [AATSR]

**Line 507+: would it be more informative to show the seasonal cycle for each hemisphere separately?**

This is a good point. We will plot the seasonal cycle for each hemisphere and revise the relevant discussion in the text.

**Page 41, Figure 13: the caption suggests that this is the difference between NEWSOA and two different observations rather than NEWSOA – OLDSOA as the legend indicates, can you correct the caption / legend?**

As stated above.

---

## Author Response (AR2)

Hello

Here are our production files for the manuscript. Below you can see the minor corrections we have made.

Best Regards,
Tommi Bergaman

Typo fixes:

Title: 1.12 changed to 1.2
Section title for 2.5.2: Number concentrations onat the surface -> Number concentrations at the surface

Figure14 c Legend was on the bottom left corner obscuring the lines, therefore it is moved to upper left corner.
Figure 14 caption:
Logarithmic R was removed from the figure during review, so the "(logarithmic R in parentheses) " was removed from the caption.

In data and code availability section following was added:
"The aerosol size distribution datasets are obtained from ACTRIS database https://actris.nilu.no (last access: 14.3.2016)."
Simulation data location has been changed due to added data files from extra simulations done during review:
https://doi.org/10.23729/b1eb8b20-83c4-4dc6-a1fd-05dc13fe42c ->
https://doi.org/10.23729/d7aee953-75f0-41eb-ba50-9b942d6215d3

Acknowledgements following corrections were made:
 Supported -> supported
After (CRESCENDO) added following
"and under grant agreement no. 821205, Constrained aerosol forcing for improved climate projections (FORCeS). RM acknowledges funding from Swedish Research Council Formas project CoBACCa (no. 2018-01745)"